# Boundary-Value PDEs Meet Higher-Order Differential Topology-aware GNNs

**Yunfeng Liao    Yangxin Wu    Xiucheng Li**[*]
School of Computer Science and Technology, Harbin Institute of Technology (Shenzhen)
`lixiucheng@hit.edu.cn`

## Abstract

Recent advances in graph neural network (GNN)-based neural operators have demonstrated significant progress in solving partial differential equations (PDEs) by effectively representing computational meshes. However, most existing approaches overlook the intrinsic physical and topological meaning of higher-order elements in the mesh, which are closely tied to differential forms. In this paper, we propose a higher-order GNN framework that incorporates higher-order interactions based on discrete and finite element exterior calculus. The time-independent boundary value problems (BVPs) in electromagnetism are instantiated to illustrate the proposed framework. It can be easily generalized to other PDEs that admit differential form formulations. Moreover, the novel physics-informed loss terms, integrated form estimators, and theoretical support are derived correspondingly. Experiments show that our proposed method outperforms the existing neural operators by large margins on BVPs in electromagnetism. Our code is available at `https://github.com/Supradax/Higher-Order-Differential-Topology-aware-GNN`.

## 1   Introduction

Solving partial differential equations (PDEs) accurately is fundamental in scientific computations. Traditional numerical solvers [1] and the emerging physics-informed neural networks [2] rely on iterative computations. This is a significant bottleneck that precludes their application in time-sensitive domains where slight inaccuracy is tolerable but speed matters, such as gaming engines and interactive simulations. To address these limitations, neural operators [3–6] propose to directly learn the mapping between initial/boundary conditions and complete PDE solutions, eliminating time-consuming iterations while maintaining the capability to handle the PDEs whose exact analytical solutions are unattainable.

Convolutional neural network-based solvers are inherently constrained to regular, grid-like domains, whereas numerical solvers typically employ meshes to represent irregular solving regions—an approach naturally aligned with graph neural networks (GNNs). This compatibility has spurred growing interest in GNNs for time-dependent physics simulations, demonstrated through applications from fabric dynamics in wind [7] and granular particle systems [8] to neural mesh refinement schemes [9]. However, solving time-independent PDEs, especially boundary value problems (BVPs), presents greater challenges due to the absence of temporal guidance (initial data) and limited feature representation. Recent work [10] attempts to apply GNNs to BVPs in electromagnetism and has shown promising results, but is still limited to relatively simple cases.

Conventional GNN-based BVP solvers primarily utilize vertex adjacency in meshes while neglecting higher-order topological elements (edges, faces, cells), despite their fundamental physics interpretations from the perspective of differential forms [11]. While vector analysis has long dominated

---

[*]Corresponding Author.

39th Conference on Neural Information Processing Systems (NeurIPS 2025).

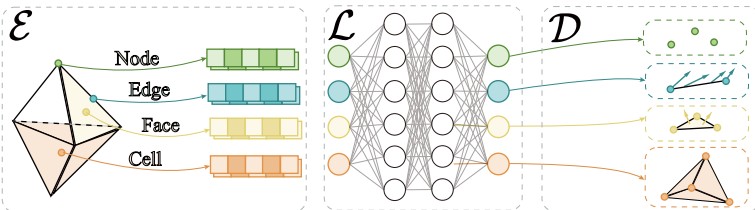

Figure 1: In the proposed DEC-HOGNN, scalar and vector fields on nodes, edges, faces and cells are encoded into $k$-simplex features, passing through HOGNN and decoded back as target fields.

physical modeling, modern physics recognizes that many vector fields are more naturally interpreted as differential forms on manifolds [12]. In electromagnetism, it reveals a hierarchical structure: potentials ($\varphi$) manifest as 0-forms, field intensities ($\mathbf{E}, \mathbf{H}$) as 1-forms, flux densities ($\mathbf{D}, \mathbf{B}$) as 2-forms and density distributions ($\rho$) as 3-forms [13]. This formalism, crystallized in the *Maxwell's House* representation of Maxwell's equations [14], treats traditional vector fields as mere proxies for underlying forms. Discrete exterior calculus [15] operationalizes this approach through De Rham mappings that represent $k$-forms as integrals over $k$-simplices [16]. Also, introducing geometric objects like differential forms allows us to naturally generalize this framework to PDEs on curved spaces [17] beyond merely incorporating topological structures [18, 19].

In this paper, we propose a higher-order GNN-based PDE solver framework by exploring the ideas in discrete exterior calculus (DEC) and finite element exterior calculus (FEEC) in a principled manner. By encoding the integrals over $k$-simplices as $k$-simplex features within higher-order GNNs (HOGNNs), which explicitly model interactions between simplices of varying dimensions, we establish a principled framework for solving form-based PDEs while preserving the topological and physical structure inherent to the problem domain. The resulting framework is dubbed DEC-HOGNN and illustrated in Figure 1. DEC-HOGNN enjoys better physical interpretation from the differential form perspective and can be extended to higher-dimension cases naturally. Our main contributions are summarized as follows: 1) We design a differential topology-aware HOGNN, which naturally encodes and decodes PDE operators based on the principles of DEC and FEEC. 2) Various physics-informed loss terms are derived under DEC-HOGNN, including the boundary condition ones, which can enable solving the boundary-value PDEs more effectively. 3) The universal approximation property in solving Poisson problems (electrostatics and magnetostatics) is presented, and the performance excellence is demonstrated via empirical experiments.

## 2   Related Work

**Higher-Order GNNs**. HOGNN is an extended learning framework on generalized graphs, i.e., hypergraphs [20]. A hypergraph $G$ allows a hyperedge to contain more than two vertices [21]. HOGNN leverages the more abundant adjacencies on hypergraphs and mimics what GNN does on plain graphs via redefining various neighborhoods. If $G$ has no further decorated structures, then one can define the $k$-node-tuple adjacency [22]; while boundary, co-boundary, upper, and lower adjacencies are available when $G$ is a simplical complex [18] or a cell complex [23]. Mechanisms in GNN are mostly based on a special adjacency induced by edges and thus can be easily transplanted to hypergraphs. The graph-convolution [24], attention [25], and generalized message passing [26] of HOGNN all fall into this category. HOGNN has been well-studied in various regions, such as recommendation systems [27] and molecular predictions [28], where multi-body interactions are of significance, but beyond the expressive ability of plain graphs.

**Neural Operators**. Neural operator [3] learns function-to-function mappings mainly using data-driven loss instead of PDE-based physics loss. Its original implementation is furnished with kernel convolution of $O(n^2)$ complexity, which can be improved to $O(n \log n)$ in the spectral domain via discrete fast Fourier transform [3]. It is then extended to non-square-like regions [4] and spatial-spectral neural operators realized by wavelet transform [29–31]. This field later gradually shifts towards Transformer architectures with PDE-compatible attentions, from GNOT [5] employing boundary-aware cross-attention to Transolver [6] using attentions among domain slices.

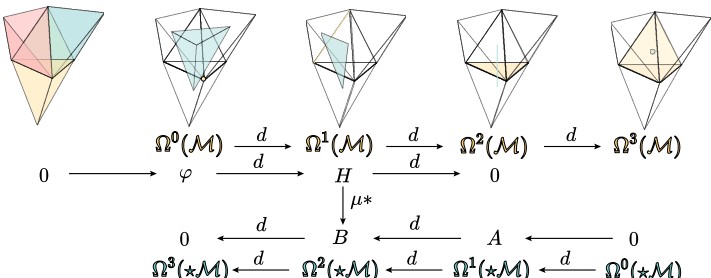

Figure 2: The primal manifold $\mathcal{M}$ has 4 tetrahedrals. The $i$-cell $\sigma_i \in \mathcal{M}$ ($i \in \{0, 1, 2, 3\}$) and its dual cell $\star\sigma_i$ are illustrated, associated with specific magnetic quantities. Identify the scalar potential $\varphi$ as 0-form, vector potential $\mathbf{A}$ and magnetic intensity $\mathbf{H}$ as 1-form and magnetic flux density $\mathbf{B}$ as 2-form, and the relations $\mathbf{H} = \nabla\varphi, \mathbf{B} = \nabla \times \mathbf{A}, \nabla \times \mathbf{H} = \mathbf{0}, \nabla \cdot \mathbf{B} = 0$ are encoded into exterior derivative $d$ and its property $d^2 = 0$. The electric case is similar.

**GNN-based PDE Solvers**. GNN-based neural operators have been studied in time-evolving mesh-based and particle-based physics simulations [7, 8]. The particles appear in the form of point clouds, and adjacency is built on local neighborhoods, in which equivariance is introduced to enhance model performance, such as subequivariance [32] and IsoGCN [33], an equivariant data-driven neural differential operator. To align with Neumann boundary in PDEs, NIsoGCN [34] further introduces a Neumann term into the differential kernel. This category requires data on fields and their differentials, but the latter is often intractable. It is found that message passing in GNNs can represent various numerical methods for time-dependent PDEs [35] and aligns with finite volume methods to achieve local mass conservation [36]. In addition, GNN-based approaches for BVPs [10] and inverse problems [37] are also explored.

## 3 Preliminary

**De-Rham Mapping**. In algebraic topology, a simplex chain complex $\mathcal{C}(X)$ on a topological space $X$ is a graded vector space of $k$-order simplices $\mathcal{C}_k(X)$, decorated with the boundary operator $\partial$. A $k$-cochain $\sigma^k \in \mathcal{C}^k(X) : \mathcal{C}_k(X) \to \mathbb{R}$ is a real-valued function of $\sigma_k \in \mathcal{C}_k(X)$. If there exists a diffeomorphism $\varphi$ between $X$ and a smooth manifold $\mathcal{M}$, then for any $k$-form $\omega$, we obtain a covariant functor mapping from $\mathcal{C}_k(X)$ to $\mathcal{C}^k(X)$, namely, De-Rham mapping [16]:

$$\mathcal{F} : \mathcal{C}_k(X) \to \mathcal{C}^k(X), \sigma_k \mapsto \int_{\varphi(\sigma_k)} \omega. \tag{1}$$

**DEC and FEEC**. *Discrete Exterior Calculus* (DEC) offers a comprehensive and differential topology-preserving toolkit to discretize operators on manifolds. In contrast to Graph Calculus [38], which treats a discretized manifold as a graph—at the cost of losing essential differential properties and thereby introducing inaccuracies—DEC maintains differential topology properties by working with integrations. Specifically, it characterizes a $k$-form $\omega \in \Omega^k(\mathcal{M})$ on a discrete manifold $\mathcal{M}$ through its integral over every $k$-simplex [16]. It yields high accuracy in applications where differential information matters, and is widely used in computational physics [39–42]. Another benefit of DEC is that it allows for processing manifolds from a dual perspective. For instance, the Hodge star $*^k : \Omega^k(\mathcal{M}) \to \Omega^{n-k}(\mathcal{M})$ on $\mathcal{M}$ gives the important constitution relation between $\mathbf{B}$ and $\mathbf{H}$ in electromagnetism. Even if $*^k\omega$ is intractable in the discrete case, we can still estimate its integral on the dual manifold $\star\mathcal{M}$ (shown in Figure 2) without a priori on the metric [15]. *Finite Element Exterior Calculus* (FEEC) generalizes the Finite Element Method (FEM) and is a numerical implementation of Galerkin methods. FEM is in essence an interpolation method with nodal functions, a.k.a., Lagrangian element [1]. However, such node-wise interpolation is inherently incompatible with differential operators such as div and curl, whereas the adoption of higher-order finite elements in FEEC enables the exact representation of these operators in the integral sense via straightforward linear combinations [14].

**Whitney Element**. Whitney elements are a type of finite elements used in FEEC and DEC. They provide a way to approximate differential forms on a discretized manifold that respects the geometry

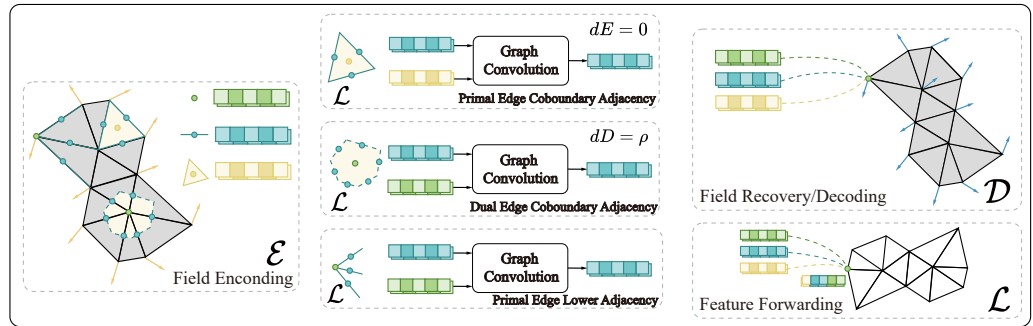

Figure 3: The illustration of DEC-HOGNN for 2D-electrostatics BVPs.

and topology of the manifold. Let $w$ and $\mathbf{w}$ denote scalar and vector fields, respectively. Given a tetrahedralized mesh representation of a manifold $\mathcal{M}$ with $\dim \mathcal{M} = 3$, let $[i, j]$ denote the edge containing $v_i, v_j$ and so are $[i, j, k], [i, j, k, l]$ for face and cell, respectively. The canonical node elements $\{w_i : w_i(v_j) = \delta_{ij}\}$ are functions on $\mathcal{M}$, in which $\delta_{ij}$ is the Kronecker delta. Let $W^0(\mathcal{M}), W^1(\mathcal{M}), W^2(\mathcal{M})$, and $W^3(\mathcal{M})$ denote the function spaces spanned by node elements $w_i$, edge elements $\mathbf{w}_{[i,j]}$, face elements $\mathbf{w}_{[i,j,k]}$, and cell elements $w_{[i,j,k,l]}$ (scalar fields), respectively. Then the integral of $\mathbf{w}_{[i,j]}$ along $[i, j]$, the flux of $\mathbf{w}_{[i,j,k]}$ through face $[i, j, k]$, and the volume integral of $w_{[i,j,k,l]}$ within tetrahedral $[i, j, k, l]$ all equal to 1 [14]. The element definitions are as follows:

$$\mathbf{w}_{[i,j]} := w_i \nabla w_j - w_j \nabla w_i \tag{2}$$

$$\mathbf{w}_{[i,j,k]} := 2(w_i(\nabla w_j \times \nabla w_k) + w_j(\nabla w_k \times \nabla w_i) + w_k(\nabla w_i \times \nabla w_j)) \tag{3}$$

$$w_{[i,j,k,l]} := 6 \sum_{\text{cyc}} w_i(\nabla w_j \times \nabla w_k) \cdot \nabla w_l = \chi_{\mathbf{x} \in [i,j,k,l]} / \text{vol}([i, j, k, l]) \tag{4}$$

in which $w_i, w_j, w_k, w_l \in W^0(\mathcal{M})$. For ease of reference, the notation frequently used throughout the paper is summarized in Table 5 in Appendix A.

## 4 Methodology

**Problem Setup and Motivation**. In this work, we focus on PDEs that admit differential form formulation (many important PDEs in physics and engineering can be formulated using differential forms, such as Maxwell's, Navier-Stokes, Yang-Mills Equations, etc.). Given PDEs (often formulated in vector fields) on a discrete manifold $\mathcal{M}$, we aim to learn a neural operator $\mathcal{G}_\Theta$ that takes as inputs the observed scalar fields $\{s_i\}$ and vector fields $\{\mathbf{v}_i\}$, and outputs the target vector field $\mathbf{u}$ of interest:

$$\mathbf{u} = \mathcal{G}_\Theta(\{s_1, ..., s_m\}, \{\mathbf{v}_1, ..., \mathbf{v}_n\}, \mathcal{M}). \tag{5}$$

In comparison to the vector field formulation, the differential-form characterization of PDEs is coordinate-independent and explicitly reveals the geometric and topological aspects of the spaces. Motivated by this, we propose to transform the vector fields into the differential form formulation of the PDEs, which enables us to develop the neural operators by combining the principles in DEC and FEEC. The benefits are threefold: 1) they permit exploring the higher-order topological elements to facilitate PDE solving, 2) developing various physics-consistent losses, especially for the boundary-value problems, and 3) differential operators can be interpreted as simple linear combinations in DEC and FEEC, aligning with the higher-order message passing framework. Once the PDEs are solved, we then translate the results into the vector field language for downstream analysis.

**Method Overview**. Since differential $k$-forms can be represented as integrals on $k$-dimensional elements and further identified as higher-order element features, we introduce physics-informed higher-order interactions and aggregations into existing HOGNN frameworks in Section 4.1. Nevertheless, the initial input and expected output are often vectors instead of forms in practice. To circumvent this, we propose a proper encoder-decoder to enable consistent transformations in Section 4.2. In a nutshell, the input vectors are encoded into forms (higher-order features), processed by

HOGNN, and then decoded back into vectors, as illustrated in Figure 1. The physics-constrained loss and the universal approximation property of the proposed method are presented in Section 4.3.

As an illustrative example, we instantiate our framework by solving the classical BVPs in electromagnetism, and the proposed method can be easily generalized to other PDEs that can be formulated using differential forms. Recall that a typical Neumann electrostatic BVP is

$$\nabla \cdot \mathbf{D} = \rho, \nabla \times \mathbf{E} = \mathbf{0} \text{ in } \Omega; \ \mathbf{E} = \mathbf{E}_0, \mathbf{D} = \mathbf{D}_0 \text{ on } \partial\Omega; \ \mathbf{D} = \varepsilon_i \cdot \mathbf{E} \text{ in } \Omega_i, \tag{6}$$

in which $\rho, \mathbf{D}, \mathbf{E}$ are the charge density, displacement field, and electric field, respectively, whereas $\{\Omega_i\}$ is a partition of the domain $\Omega$. The permittivity tensor $\varepsilon_i$ can vary in different $\Omega_i$ consisting of different materials. In this case, the learned operator $\mathcal{G}_\Theta$ will take as input $(\{\rho\}, \{\mathbf{E}_0, \mathbf{D}_0\}, \Omega)$ and produce the complete fields $(\mathbf{E}, \mathbf{D})$. To this end, we rewrite Eq. 6 in its differential form formulation.

$$dD = \rho, dE = 0 \text{ in } \Omega; \ E = E_0, D = D_0 \text{ on } \partial\Omega; \ D = \varepsilon *^1 E \text{ in } \Omega_i, \tag{7}$$

in which $d\cdot$ denotes the exterior derivative and $D, E$ are the corresponding $k$-forms. The input scalar field $\rho$ and masked vector fields $\mathbf{E}, \mathbf{D}$ are first encoded into integrated forms in the form of vertex, edge and face features. Three types of edge adjacencies are used in the higher-order GNN, which give important physics interpretations (as shown in Figure 3): the curl-free property $dE = 0$ is implicitly included in the primal edge co-boundary adjacency, while Gauss's Law $dD = \rho$ is involved in the dual edge co-boundary adjacency. These GNN layers can be stacked sequentially, and the aggregated features can be either decoded back as complete $\mathbf{E}, \mathbf{D}$ or forwarded to other networks.

## 4.1 Physics-Informed Higher-Order Interactions

In light of DEC, we identify the potential on node, the circulation along an edge, the flux through a face, and the mass within a cell as the node, edge, face, and cell features, respectively. Given a discrete manifold $\mathcal{M}$ and its dual $\star\mathcal{M}$, since both of them are cell complexes, an element $c$ onward have four types of neighborhood: boundary $\mathcal{B}(c) := \partial c$ and co-boundary $\mathcal{C}(c) := \{d : c \in \mathcal{B}(d)\}$, upper adjacency $\mathcal{N}_\uparrow(c) := \{d : \exists\delta, \{c, d\} \subset \mathcal{B}(\delta)\}$ and lower adjacency $\mathcal{N}_\downarrow(c) := \{d : \exists\delta, \{c, d\} \subset \mathcal{C}(\delta)\}$. Following the paradigm of [26], the element feature $\mathbf{h}'_c$ of $c$ is updated by its original feature $\mathbf{h}_c$ and the aggregated messages from four neighborhoods $\mathcal{B}(c), \mathcal{C}(c), \mathcal{N}_\uparrow(c), \mathcal{N}_\downarrow(c)$ via:

$$\mathbf{h}'_c = \varphi\left(\mathbf{h}_c, \mathbf{m}_c^{\mathcal{B}}, \mathbf{m}_c^{\mathcal{C}}, \mathbf{m}_c^{\mathcal{N}_\downarrow}, \mathbf{m}_c^{\mathcal{N}_\uparrow}\right), \quad c \in \mathcal{M}, \tag{8}$$

$$\mathbf{m}_c^{\mathcal{B}} = \text{Aggregate}_{d\in\mathcal{B}(c)} \, \varphi_{\mathcal{B}}(\mathbf{h}_c, \mathbf{h}_d), \tag{9}$$

$$\mathbf{m}_c^{\mathcal{C}} = \text{Aggregate}_{d\in\mathcal{C}(c)} \, \varphi_{\mathcal{C}}(\mathbf{h}_c, \mathbf{h}_d), \tag{10}$$

$$\mathbf{m}_c^{\mathcal{N}_\uparrow} = \text{Aggregate}_{d\in\mathcal{N}_\uparrow(c), \, \delta\in\mathcal{C}(c)\cap\mathcal{C}(d)} \, \varphi_{\mathcal{N}_\uparrow}(\mathbf{h}_c, \mathbf{h}_d, \mathbf{h}_\delta), \tag{11}$$

$$\mathbf{m}_c^{\mathcal{N}_\downarrow} = \text{Aggregate}_{d\in\mathcal{N}_\downarrow(c), \, \delta\in\mathcal{B}(c)\cap\mathcal{B}(d)} \, \varphi_{\mathcal{N}_\downarrow}(\mathbf{h}_c, \mathbf{h}_d, \mathbf{h}_\delta). \tag{12}$$

Since forms in a PDE can be defined on both $\mathcal{M}$ and $\star\mathcal{M}$ (e.g., $E$ and $D$ in Eq. 7), we also propose a higher-order MPNN on the dual manifold $\star\mathcal{M}$ for dual forms by the fact:

$$\mathcal{B}(\star c) = \{\star d : d \in \mathcal{C}(c)\}, \quad \mathcal{C}(\star c) = \{\star d : d \in \mathcal{B}(c)\},$$
$$\mathcal{N}_\uparrow(\star c) = \{\star d : d \in \mathcal{N}_\downarrow(c)\}, \quad \mathcal{N}_\downarrow(\star c) = \{\star d : d \in \mathcal{N}_\uparrow(c)\}.$$

Usually, not all neighborhoods and elements will be used in higher-order MPNN due to computational complexity and the absence of features on corresponding elements. For instance, 2D-electrostatics BVPs only involve $\mathbf{E}, \mathbf{D}, \rho$ and thus only adjacencies about edges and faces are considered. In recent studies, there are various ways to implement the message passing on four different adjacencies, including generalized convolution, attention, and Transformer. We list several candidates of the adjacency layer backbones in Table 1 and will study their impact on model efficacy in Section 5.2.

In Table 1, $L^{(k)} := D - A = 2D - II^\top$ is the higher-order Laplacian [21], defined by indicator matrix $I_{ij} := \chi_{\sigma_i \in \mathcal{N}_\downarrow(\sigma_j)}$ and diagonal degree matrix $D$; $\tilde{L}$ is the Laplacian of an extended graph $G = (\mathcal{C}_k \cup \mathcal{C}_{k+1}, \{(c, d) : c \in \mathcal{C}_k, d \in \mathcal{C}(c)\})$; $\mathbf{h}_{c,d,\delta}$ is the concatenation of $\mathbf{h}_c, \mathbf{h}_d, \mathbf{h}_\delta$ and $\mathbf{h}_{c,d}$ is likewise. In our implementation, the relative orientation $\text{sign}(c, d)$ between two elements is further considered, via replacing $\mathbf{h}_{c,d}$ by $\text{sign}(c, d)\mathbf{h}_{c,d}$.

**Consistency with Conservation Law**. It is beneficial for solving BVPs by introducing higher-order interactions, because the interactions have meaningful physics interpretations, e.g., the sum of edge

Table 1: Possible implementations for co-boundary/lower adjacency layers.

| Layer | Co-boundary and lower adjacencies |
|-------|-----------------------------------|
| Convolution | $\mathbf{h}_c'^{(k)} = \sigma\big(|\mathcal{N}_\downarrow(c)|^{-\frac{1}{2}} \sum\limits_{(d,\delta)\in\mathcal{N}_\downarrow(c)} \big(L^{(k)}\mathbf{h}^{(k)}\Theta^{(k)}\big)_d + \big(L^{(k+1)}\mathbf{h}^{(k+1)}\Theta^{(k+1)}\big)_\delta\big)$ |
| | $\mathbf{h}_c'^{(k)} = \sigma\big(|\mathcal{C}(c)|^{-\frac{1}{2}} \sum\limits_{d\in\mathcal{C}(c)} \big(\tilde{L}^{(k)}\mathbf{h}^{(k)}\Theta^{(k)}\big)_d\big)$ |
| Attention | $\mathbf{h}_c' = \sigma(W_C\mathbf{h}_c + \sum\limits_{(d,\delta)\in\mathcal{N}_\downarrow(c)} \mathrm{softmax}_{\mathcal{N}_\downarrow(c)}(\alpha_{d\delta})W_N\mathbf{h}_{c,d,\delta}), \alpha_{d\delta} = W_A\mathbf{h}_{c,d,\delta}$ |
| | $\mathbf{h}_c' = \sigma(W_C\mathbf{h}_c + \sum\limits_{d\in\mathcal{C}(c)} \mathrm{softmax}_{\mathcal{C}(c)}(\alpha_d)W_N\mathbf{h}_{c,d}), \alpha_d = W_A\mathbf{h}_{c,d}$ |
| Transformer | $\mathbf{h}_c' = \sigma(W_C\mathbf{h}_c + \sum\limits_{(d,\delta)\in\mathcal{N}_\downarrow(c)} \mathrm{softmax}_{\mathcal{N}_\downarrow(c)}(|\mathcal{N}_\downarrow(c)|^{-\frac{1}{2}}\mathbf{q}_c^\top\mathbf{k}_d)W_N\mathbf{h}_d)$ |
| | $\mathbf{h}_c' = \sigma(W_C\mathbf{h}_c + \sum\limits_{d\in\mathcal{C}(c)} \mathrm{softmax}_{\mathcal{C}(c)}(|\mathcal{C}(c)|^{-\frac{1}{2}}\mathbf{q}_c^\top\mathbf{k}_d)W_N\mathbf{h}_d)$ |

features around a face corresponds to the vorticity, while the sum of face features around a cell indicates the divergence at that location. This enables us to preserve structural conservation discretely by using integrated differential forms. Moreover, prior work also shows that adding conservation regularization can boost the performance of PDE solvers. For instance, a conservative GNN solver for 2D fluid dynamics [36] employs message passing based on face lower adjacency in the HOGNN framework and achieves local mass conservation (divergence-free) through asymmetric aggregation. We can realize more conservation constraints with DEC, e.g., enforcing vorticity conservation ($\nabla \times \mathbf{E} = 0$) and divergence conservation ($\nabla \cdot \mathbf{B} = 0, \nabla \cdot \mathbf{D} = \rho$) in electromagnetism.

**Consistency with Electromagnetic Constitution Law**. As a side-product, we can estimate the electric and magnetic permeability $\varepsilon$ and $\mu$ of the medium while solving BVPs. The constitution law $\mathbf{B} = \mu\mathbf{H}, \mathbf{D} = \varepsilon\mathbf{E}$ can be equivalently interpreted by Hodge star: $B = \mu *^1 H, D = \varepsilon *^1 E$. It allows us to estimate the $\mu, \varepsilon$ along different directions via the definition in DEC:

$$\int_{\star\sigma_k} *^k\omega := \frac{\mathrm{vol}\,(\star\sigma_k)}{\mathrm{vol}\,(\sigma_k)} \int_{\sigma_k} \omega, \quad \omega \in \Omega^k(\mathcal{M}). \tag{13}$$

## 4.2 Encoder-Decoder between Vector Fields and Forms

**Encoder**. The encoder aims to transform vector fields into higher-order element features (integrated forms). In 3D-cases, a 1-form $\omega_1$ and a 2-form $\omega_2$ can be realized by *vector proxies* $\mathbf{u}_1$ and $\mathbf{u}_2$, with the aid of a unit tangent vector $\mathbf{t}$ on an edge $\sigma_1$ and a unit normal vector $\mathbf{n}$ on a face $\sigma_2$, respectively; while a 3-form $\omega_3$ can be identified as a scalar field $u_3 : \mathbb{R}^3 \to \mathbb{R}$, and the integrated form is the usual volume integral. More formally,

$$\int_{\sigma_1} \omega_1 := \int_{\sigma_1} \mathbf{u}_1 \cdot \mathbf{t}ds, \quad \int_{\sigma_2} \omega_2 := \int_{\sigma_2} \mathbf{u}_2 \cdot \mathbf{n}dS, \quad \int_{\sigma_3} \omega_3 := \int_{\sigma_3} u_3 dV. \tag{14}$$

Given the samples $\{\mathbf{x}_j\}$ observed on $\sigma_i$ ($i = 1, 2, 3$), the integrals on the right-hand sides of Eq. 14 can be estimated to yield the integrated forms numerically with either the Monte Carlo method or canonical quadrature rules on simplices (e.g., cubic quadrature on a triangle [43]). Similarly, the 2D case is given as follows and can be estimated similarly:

$$\int_{\sigma_1} w_1 = \int_{\sigma_1} \mathbf{u}_1 \cdot \mathbf{t}ds, \quad \int_{\sigma_2} w_2 := \int_{\sigma_2} u_2 dxdy. \tag{15}$$

In DEC, the differential forms in the primal manifold and dual manifold are related by the Hodge star $*^1$ (e.g., $E \in \Omega^1(\mathcal{M})$ and $D \in \Omega^2(\star\mathcal{M})$ in Eq. 7). Since we only have observations at primal nodes, it is simpler to estimate the integrated 2-forms on the primal manifold. However, DEC requires the integrals on the dual manifold, which poses a challenge for implementation. To sidestep this, we derive an approximation of the integrated dual forms based on the integrated primal forms, presented in Theorem 1. The details of the theorem can be found in Appendix C. Intuitively, the theorem offers us a simpler way to calculate the integrated dual forms.

**Theorem 1** (Non-Cycle Forms Estimation). *For a smooth k-form $\omega$ that is not a cycle defined on a bounded region, i.e., $d\omega$ does not vanish identically in any k-simplex $\sigma_k$, one can estimate the integrated dual forms $\{\int_{\star\sigma_{n-k}} \omega : \sigma_{n-k} \in \mathcal{C}_{n-k}(X)\}$ by using the integrated primal forms $\{\int_{\sigma_k} \omega : \sigma_k \in \mathcal{C}_k(X)\}$ up to accuracy $O(\varepsilon^{k+1})$ in which $\varepsilon := \sup_{\sigma_k \in \mathcal{C}_k(X)} \operatorname{diam} \sigma_k$.*

**Decoder**. The decoder intends to recover the vector fields from the derived integrated forms. The principle is that the integral of a proper vector field proxy on a simplex should equal the integrated form onward. To find the proxy at $v_i$ for a 1-form (e.g., $E$ in Eq. 7) along edge $[v_i, v_j]$, note that $h_{[v_i,v_j]}$—the final edge feature output by DEC-HOGNN—indicates the circulation of the proxy and the circulation of $\mathbf{w}_{[v_i,v_j]}$ in Eq. 2 along edge $[v_i, v_j]$ is 1, and thus the Whitney electric field contributed by edge $[v_i, v_j]$ at $v_i$ is set to $h_{[v_i,v_j]}\mathbf{w}_{[v_i,v_j]}$. The contributions to $v_i$ from different edges can be aggregated by average pooling or attention-style weighted sum. Similarly, for the proxies of 2-forms (e.g., $D$ in Eq. 7) with final face feature $h_{[v_i,v_j,v_k]}$ and 3-forms (e.g., $\rho$ in Eq. 7) with final cell feature $h_{[v_i,v_j,v_k,v_l]}$, the face and cell contribution are $h_{[v_i,v_j,v_k]}\mathbf{w}_{[v_i,v_j,v_k]}$ and $h_{[v_i,v_j,v_k,v_l]}w_{[v_i,v_j,v_k,v_l]}$ respectively, with $\mathbf{w}_{[v_i,v_j,v_k]}, w_{[v_i,v_j,v_k,v_l]}$ defined by Eq. 3 and Eq. 4.

**Remark**. The interpolation nature of the decoding approach offers a continuous field not only defined on the nodes but also in the cells. Though other GNN operators can also estimate the inner fields by multi-linear interpolation based on values at nodes, such methods can destroy certain important physical properties compared with Whitney elements. For instance, the reflection law in electrodynamics elucidates that the tangent component of $\mathbf{E}$ on the medium interface is continuous while the normal component is not. With prior knowledge of the interface positions, one can choose to average out contributions from each side respectively and obtain two different $\mathbf{E}$'s before and after reflection, which is consistent with physics on the interface, while a usual interpolation based on node-wise data cannot. Lastly, we present the theoretical analysis in Theorem 2, demonstrating that our encoding and decoding scheme preserves sufficient information for scalar and vector fields on a sufficiently fine mesh.

**Theorem 2** (Proper Encoder-Decoder). *Given a uniform partition $\{[x_i, x_{i+1}] : 1 \leq i \leq 2^N\}$ on the unit interval, i.e., $0 = x_1 < x_2... < x_{2^N+1} = 1$, the encoding operator $\mathcal{E}_N : W^{1,2}([0,1]) \to \mathbb{R}^{2^N}$ is defined as the $2^N$ integrals on each $[x_i, x_{i+1}]$ and the decoding operator $\mathcal{D}_N : \mathbb{R}^{2^N} \to C^\infty([0,1])$ maps any $2^N$-dimensional feature $\mathbf{h}$ to a function $f(x)$ on $[0,1]$ such that $\int_{x_i}^{x_{i+1}} f(x)dx = \mathbf{h}_i$. Then for any $\varphi \in L^2([0,1])$ and $\varepsilon > 0$, there exists an integer $M > 0$ such that for all $N > M$,*

$$||\varphi - \mathcal{D}_N \circ \mathcal{E}_N(\varphi)||_{L^2} < \varepsilon. \tag{16}$$

*By definition, we have $\mathcal{E}_N \circ \mathcal{D}_N = \operatorname{id}$.*

### 4.3 Physics-Informed Loss and Universal Approximation Property

**Physics-Informed Loss**. Purely data-driven methods are likely to go against physics. Hence, PINN introduces physics-constrained loss as regularization to guide models to learn beyond the data resolution [44]. In PINN, the PDE residual is usually estimated by point-wise sampling and auto-differentiation. This is incompatible with GNNs since vertices are discrete, making it hard to enforce physics-consistency constraints, e.g., constraining a field to be curl-free [10]. But in integrated forms, one can interpret a differential version of PDEs into the integral version. For instance, magnetic flux density $\mathbf{B}$ is always divergence-free, inducing a DEC-version constraint:

$$dB = 0 \Leftrightarrow \nabla \cdot \mathbf{B} = 0 \Rightarrow \int_\Omega \nabla \cdot \mathbf{B} = \int_{\partial\Omega} \mathbf{B} \cdot \mathbf{n} = \sum_{\sigma_2 \in \partial\Omega} \int_{\sigma_2} \mathbf{B} \cdot \mathbf{n}_{\sigma_2} = \sum_{\sigma_2 \in \partial\Omega} \int_{\sigma_2} B = 0. \tag{17}$$

In our proposed framework, $\int_{\sigma_2} B$ is indeed a feature of face $\sigma_2$. Therefore, we can introduce physics-informed loss without sampling and differentiation, but by simply summing up corresponding higher-order features. More physics-informed loss terms are derived in Appendix B, covering the divergence, vorticity, and boundary conditions.

**Universal Approximation Property**. Theorem 3 shows the universal approximation ability of DEC-HOGNN in solving the Poisson problems. The proof is available in Appendix D.

**Theorem 3** (Universal Approximation Property). *Let $H^2(\Omega)$ denote the Hilbert space on a bounded closed region $\Omega$ with $C^1$-boundary, and $\mathcal{P}_i := \mathcal{D}_i \circ \mathcal{E}_i$ be the encoder-decoder projection with*

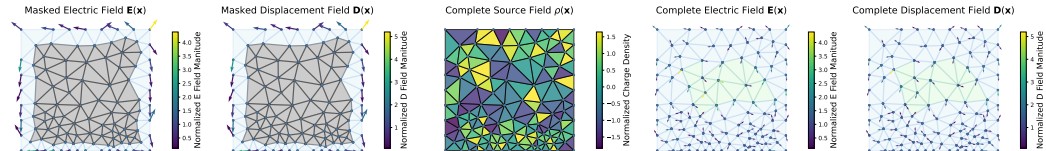

Figure 4: In the electrostatic field task, the model is required to output complete fields shown in the right-side two panels based on the left-side three panels.

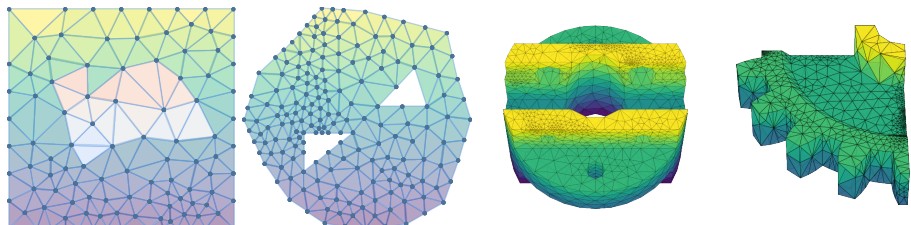

Figure 5: Models are compared on various settings. From left to right: 2D electrostatics BVP on a square 2D-mesh with two kinds of medium; 2D magnetostatics on an irregular holed mesh; 3D electrostatics and magnetostatics on a 3D socket model and a quarter-gear model, respectively.

*resolution $i$. Given a simplex-partition $\{X_\alpha\}$ of $\Omega$ with measure-zero intersections, for any $f \in V \cup \bigcup_{i=1}^{\infty} \mathcal{P}_i V, V := H^2([0,1]^n) \cap \{f : ||\nabla f||_\infty < M\}$, there exists a network in the form of $\mathcal{D} \circ \mathcal{L}_N ... \circ \mathcal{L}_1 \circ \mathcal{E}$ with infinite neurons to solve the Dirichlet Poission problem on $\Omega$, such that:*

$$\lim_{\sup diamX_\alpha \to 0} ||\mathcal{D} \circ \mathcal{L}_N ... \circ \mathcal{L}_1 \circ \mathcal{E}(f,g) - u||_{L^2} = 0, \tag{18}$$

*in which $\Delta u(\mathbf{x}) = f(\mathbf{x}), \mathbf{x} \in \Omega; u(\mathbf{x}) = g(x), \mathbf{x} \in \partial\Omega$.*

As a corollary, note that $\mathbf{E} = \nabla u$ and the experiment setting below is equivalent to solving a Neumann boundary condition in a linear medium, the universal approximation property also rings true by a similar proof in Appendix D via single-layer potential method.

## 5 Experiment

### 5.1 Experiments and Benchmarks

**Experiment Tasks**. We assess the performance of the proposed model on 2D electric and magnetic boundary value problems. In the electrostatic case, the system is governed by a potential $\varphi$:

$$\Delta\varphi(\mathbf{x}) = \rho(\mathbf{x}), \mathbf{x} \in \Omega; \nabla\varphi(\mathbf{x}) = \mathbf{u}(\mathbf{x}), \mathbf{x} \in \partial\Omega,$$

which gives the electric intensity $\mathbf{E}$ and electric displacement $\mathbf{D}$. In our settings, the domain $\Omega$ is partitioned into two regions $\Omega_1, \Omega_2$ with measure-zero intersection. $\Omega_1$ has isotropic permeability $\varepsilon_1 I$ while $\Omega_2$ has a non-isotropic linear one $\varepsilon_2$, as shown in the blue and green-colored regions in Figure 4, and

$$\mathbf{D}(\mathbf{x}) = \varepsilon_i \mathbf{E}(\mathbf{x}), \mathbf{x} \in \Omega_i, i = 1, 2.$$

Let $H^2(\Omega)$ be the function space on $\Omega$ with $L^2$ derivatives, i.e., $H^2(\Omega) = \{f : ||f||_2^2 + ||\nabla f||_2^2 < \infty\}$ and $\mathbb{H}^2(\Omega)$ the vector-valued function space on $\Omega$ with $H^2(\Omega)$ components. The operator $\mathcal{G} : H^2(\Omega) \times \mathbb{H}^2(\partial\Omega)^2 \to \mathbb{H}^2(\Omega)^2, (\rho, \mathbf{E} \cdot \chi_{\mathbf{x} \in \partial\Omega}, \mathbf{D} \cdot \chi_{\mathbf{x} \in \partial\Omega}) \mapsto (\mathbf{E}, \mathbf{D})$ recovers the field $\mathbf{E}, \mathbf{D}$ on $\Omega$ based on prior knowledge on the boundary field data $(\mathbf{E}, \mathbf{D})$ on $\partial\Omega$ and the charge density distribution $\rho$ on $\Omega$, as shown in Figure 4.

**Dataset Generation**. Both 2D and 3D meshes are adopted as illustrated in Figure 5. The electrostatic data is obtained by FEM-based electromagnetism PDE solvers. $\Omega$ is partitioned into many triangles $X_i$ with measure-zero intersections. The charge density $\rho_i$ in each $X_i$ is randomly assigned following a uniform distribution. The PDEs are then solved in a large enough vacuum region with Neumann boundary conditions. $\mathbf{E}, \mathbf{D}$ on boundary vertices and $\rho$ on all vertices are sampled as input while

Table 2: Performance of different neural operators in terms of MSE. The bold and underscored numbers indicate the best and second-best in each column, respectively.

| Model | 2D Electrostatics | 2D Magnetostatics | 3D Electrostatics | 3D Magnetostatics |
|---|---|---|---|---|
| DeepONet | 1.866±0.025 | 1.438±0.021 | 0.842±0.094 | 0.483±0.001 |
| MKGN | 1.960±0.097 | 2.324±0.209 | 0.793±0.043 | 1.382±0.030 |
| Galerkin-Type | 1.209±0.028 | 1.895±0.144 | 1.176±0.030 | **0.120±0.003** |
| GNOT | 2.064±0.052 | 1.142±0.149 | 7.266±0.014 | 4.480±0.543 |
| Transolver | 2.751±0.009 | 7.432±0.143 | 8.249±0.009 | 7.089±0.460 |
| GCN-based | 1.362±0.015 | 1.224±0.026 | 1.986±0.892 | 0.182±0.009 |
| GAT-based | 1.360±0.014 | 1.755±0.040 | 1.029±0.049 | 1.222±0.021 |
| Graph UNet-based | 1.917±0.024 | 1.283±0.004 | 0.809±0.061 | 0.375±0.006 |
| GT-based | 0.990±0.031 | 1.405±0.077 | 0.244±0.025 | 0.171±0.023 |
| DEC-HOGNN (**Ours**) | **0.623±0.058** | **0.875±0.052** | **0.195±0.005** | 0.158±0.008 |

Table 3: Test loss and performance drop of different variants against the entire models.

| Interactions | Lower adjacency | PD | P | D | None | PD | PD | PD | None |
|---|---|---|---|---|---|---|---|---|---|
| | Co-boundary | PD | PD | PD | PD | D | P | None | None |
| Test loss (mean square error) | | 0.652 | 0.676 | 0.678 | 0.719 | 0.954 | 0.959 | 1.244 | 2.077 |
| Performance drop (%) | | 0.00 | 3.81 | 4.12 | 10.34 | 46.35 | 47.23 | 90.98 | 218.87 |

$\mathbf{E}, \mathbf{D}$ on all vertices are the target output. Scalar fields are normalized, and vector fields are shrunk with respect to the average norm to eliminate magnitude differences.

Note that the 2D magnetostatic field is also governed by a potential $A_z$ (the vector potential $\mathbf{A}$ has only one non-vanishing component $A_z$ if the 2D space is identified as the $xy$-plane). Therefore, the data generation is similar. For more details, please refer to Appendix E.

## 5.2 Numerical Results

**Baselines and Implementation Details**. To showcase the necessity of introducing specifically devised solvers for BVPs and the efficacy of the proposed method, we evaluate our model against the following general time-dependent PDE neural solvers, in which the input fields are masked accordingly: DeepONet [45], MKGN [46], Galerkin-type Attention [47], GNOT [5], and Transolver [6]; in addition, various GNN-based solvers devised particularly for BVPs are also included (as in [10]): GCN [48], GAT [49], Graph U-Net [50], and Graph-Transformer-based BVP solvers [51].

**Main Results.** Table 2 presents the MSE (mean square error) of different methods on 2D/3D electrostatics/magnetostatics BVPs. The neural operators from DeepONet to Transolver, which are mostly designed for time-dependent PDEs, yield larger errors compared with GNN-based BVP solvers. It is because many time-evolving PDE solvers like DeepONet predict an increment based on previous observations, which is unfortunately intractable in static cases. The multi-scale design in MKGN and the heterogeneous cross-attention in GNOT both suffer from the lack of features, as time-independent PDEs can be determined by the boundary conditions on boundary nodes, which takes up a minority. These make them no better than simple GNN operators in [10]. The proposed DEC-HOGNN considers implicit higher-order interactions in the governing systems and thereby outperforms usual GNN operators.

**Ablation I: On Higher-Order Interactions**. Different types of interactions can contribute variably to the performance of DEC-HOGNN. In this experiment, the full model incorporates four types of interactions, namely, primal and dual lower/co-boundary adjacencies, which are systematically ablated to evaluate their impact. The resulting variants are denoted as Primal-Dual (PD), Primal-only (P), Dual-only (D), and None. Table 3 presents the results in 2D electrostatic BVPs, which shows that message passing on edge lower adjacency brings minor enhancement while co-boundary adjacency matters much more. This result coincides with the governing PDE in electrostatics, $dE = 0, dD = \rho$, whose inducing loss terms are supposed to be computed based on both primal and dual co-boundary adjacency (Appendix B).

Table 4: The improvements of DEC-HOGNNs equipped with different higher-order graph convolutions and interactions against the vanilla GNN on the 2D-magnetostatics.

| Backbone | GCN | GAT | Graph Transformer |
|---|---|---|---|
| Both co-boundary and lower adjacency | 2.450±0.050 | 2.246±0.350 | **1.222±0.036** |
| Lower adjacency only | 3.463±0.069 | 3.491±0.081 | 1.629±0.068 |
| Co-boundary only | 2.159±0.045 | **2.161±0.030** | 1.386±0.041 |
| Vanilla GNN | **1.876±0.038** | 2.747±0.541 | 1.646±0.026 |
| Improvement (%) | -15.07 | 21.33 | 25.75 |

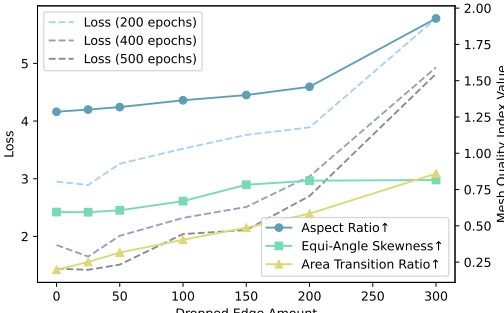 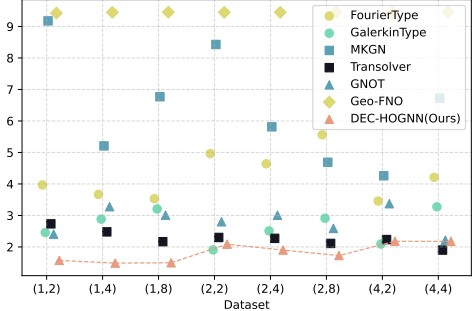

Figure 6: Left: Mesh quality and model performance of DEC-HOGNN variations as more edges are dropped; Right: Model comparison on meshes with different topological characteristics.

**Ablation II: On Different GNN Backbones**. In this experiment, we study the impact of different variants of higher-order GNNs on the efficacy of DEC-HOGNN by selecting different backbones, including GCN, GAT, and Graph Transformer (GT), on 2D electrostatic BVPs. Results in Table 4 manifest that higher-order interactions can enhance model performance (GAT and GT), which is consistent with the previous results. It further shows that adding more higher-order interactions does not necessarily imply more positive sides. For instance, the appearance of lower adjacency impedes GAT and GCN, while the co-boundary adjacency is much more beneficial for all backbones. Also, not every convolution can fit the DEC-HOGNN framework well just like the counterexample GCN.

**Ablation III: On Performance Degeneration due to Mesh Quality**. We analyze the negative impact out of mesh degradation by randomly dropping a certain amount of edges hierarchically. To measure the mesh quality quantitatively, three indicators are adopted with arrows implying degrading directions since good quality usually comes with uniform and regular elements. Further details on these tailored meshes are covered in Appendix E. The left panel of Figure 6 reflects that dropping edges from a triangularized mesh is followed by mesh degeneration. Also, minor degeneration would not affect the performance violently while a major one leads to salient performance drop. Note that it is also infeasible to adopt classical solvers using meshes with prominent quality issues. Thus these negative effects are tolerable.

**Ablation IV: On Different Topological Characteristics**. As shown in Figure 12, eight 2D magnetotastics benchmarks are used for the evaluation, which are named after their different topological properties by *(Connected Component Amount, Hole Amount)*. The right panel of Figure 6 shows that the advantage of our approach persists as the underlying topology changes.

## 6 Conclusion

In this paper, we propose a BVP solver via integrating higher-order topological interactions, which aligns with the discrete representation of differential forms, and the resulting model is aware of differential topology. Several novel physics-informed loss terms and integrated form estimators are also developed. Both theoretical analysis and experimental results demonstrate the advantages of incorporating higher-order interactions via integrated differential forms. Similar to traditional mesh-based numerical solvers, the performance of DEC-HOGNN is influenced by mesh quality, as Whitney elements may degrade on poorly shaped triangulated or tetrahedral meshes, potentially leading to convergence issues. Extending this approach to more general time-dependent PDEs and developing methods to mitigate mesh quality dependence are left for our future work.

## Acknowledgements

This work is supported by the National Natural Science Foundation of China under Grant No. 62206074 and 62472125, Guangdong Basic and Applied Basic Research Foundation under Grant No. 2025A1515012932, Shenzhen College Stability Support Plan under Grant No. GXWD20220811173233001, Shenzhen Science and Technology Program No. JCYJ20241202123503005 and ZDSYS20230626091203008.

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

Table 5: Summary of symbols and their corresponding meanings.

| Symbol | Meaning |
|---|---|
| $\Omega^k(\mathcal{M})$ | The space spanned by all $k$-form on manifold $\mathcal{M}$. |
| $\star\mathcal{M}$ | The dual manifold of the primal manifold $\mathcal{M}$. |
| $\mathcal{C}(X)$ | The simplex chain complex of topology space $X$. |
| $\mathcal{C}_k(X)$ | The set of all $k$-simplices in $\mathcal{C}(X)$. Its element is denoted by $\sigma_k$. |
| $\mathcal{C}^k(X)$ | The set of all $k$-cochain on $\mathcal{C}_k(X)$. $\forall \sigma^k \in \mathcal{C}_k(X), \sigma^k : \mathcal{C}_k(X) \to \mathbb{R}, \sigma_k \mapsto \sigma^k(\sigma_k)$. |
| $*^k$ | The Hodge star acting on k-forms. $*^k : \Omega^k(\mathcal{M}) \to \Omega^{n-k}(\mathcal{M})$. In DEC, it maps $k$-forms on the discrete primal manifold to the discrete dual manifold, i.e., $*^k : \Omega^k(\mathcal{M}) \to \Omega^{n-k}(\star\mathcal{M})$ |
| $[i_0, ..., i_n]$ | A $n$-simplex with vertices $v_{i_0}, v_{i_1}, ..., v_{i_n}$ in order. |
| $d^k$ | The exterior derivative. $d^k : \Omega^k(\mathcal{M}) \to \Omega^{k+1}(\star\mathcal{M})$. |
| $\partial^k$ | The boundary operator. $\partial^k : \mathcal{C}_k(X) \to \mathcal{C}_{k-1}(X)$. |
| $H^2(\Omega)$ | The function space on region $\Omega$ with $L^2$ derivatives. |
| $\mathbb{H}^2(\Omega)$ | The vector-valued function space on region $\Omega$, each component having $L^2$ derivative. |
| $W^k(\mathcal{M})$ | The space spanned by $k$-th order Whitney element defined on $\mathcal{M}$. |
| $\chi_A$ | Indicator function. $\chi_A = 1$ if condition $A$ is satisified and otherwise $\chi_A = 0$. |

## A   Notation

The notation used throughout the paper is summarized in Table 5.

## B   Training Loss Function Derivation

The loss below is under the assumption that the dual manifold is constructed based on circumcenters, and fields are recovered by interpolation. For brevity, we only focus on discrete manifolds in $\mathbb{R}^3$.

**Div-free and Curl-free**. $\nabla \cdot \mathbf{B} = 0 \Leftrightarrow dB = 0, B \in \Omega^2(\star\mathcal{M})$. By Stokes theorem,

$$0 = \int_{\sigma_3} dB = \int_{\partial\sigma_3} B = \sum_{\sigma_2 \prec \sigma_3} \int_{\sigma_2} B = \sum_{\sigma_2 \prec \sigma_3} \text{sign}(\sigma_2)\mathbf{h}_{\sigma_2} \tag{19}$$

in which the partial-order relation $\sigma \prec \sigma'$ implies that $\sigma$ is a face of $\sigma'$. And thus the physics-informed $L^2$ loss can be:

$$\mathcal{L}_{\text{div-free}} := \sum_{\sigma_3 \in \mathcal{M}} \left| \sum_{\sigma_2 \prec \sigma_3} \text{sign}(\sigma_2, \sigma_3)\mathbf{h}_{\sigma_2} \right|^2 \tag{20}$$

Then for $\nabla \times \mathbf{H} = 0 \Leftrightarrow dH = 0, H \in \Omega^1(\mathcal{M})$, likewise, we have:

$$\mathcal{L}_{\text{curl-free}} := \sum_{\sigma_2 \in \star\mathcal{M}} \left| \sum_{\sigma_1 \prec \sigma_2} \text{sign}(\sigma_1, \sigma_2)\mathbf{h}_{\sigma_1} \right|^2 \tag{21}$$

As for forms $\mathbf{D}, \mathbf{E}$ such that $\nabla \cdot \mathbf{D} = 0, \nabla \times \mathbf{E} = 0$, one can obtain the formula by modifying $\sigma_3 \in \mathcal{M}, \sigma_2 \in \star\mathcal{M}$ by $\sigma_3 \in \star\mathcal{M}, \sigma_2 \in \mathcal{M}$. And for brevity, we will only cover the case on $\star\mathcal{M}$ since that on $\mathcal{M}$ is almost the same. For non-divergence-free scenarios like $\nabla \cdot \mathbf{D} = \rho$, then the $L^2$ loss becomes:

$$\mathcal{L}_{\text{div}} := \sum_{\sigma_3 \in \star\mathcal{M}} \left| \rho(\sigma_3)\,\text{vol}(\sigma_3) - \sum_{\sigma_2 \prec \sigma_3} \text{sign}(\sigma_2, \sigma_3)\mathbf{h}_{\sigma_2} \right|^2 \tag{22}$$

**Boundary Condition: $\mathbf{n} \cdot \mathbf{B} = 0$.** Consider a tetrahedral $\sigma_3 := [A, B, C, D]$ with positive-oriented faces $[A, B, C], [C, D, A], [A, D, B], [B, D, C]$, as shown in Figure 7. Let $h_A$ denote the altitude to the opposite face $[B, D, C]$ in $\sigma_3$, $w_A$ the nodal function at vertex $A$, and similarly for the others.

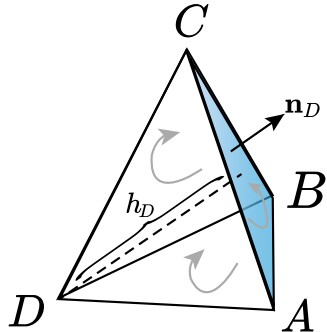

Figure 7: Tetrahedral $[A, B, C, D]$.

$\overrightarrow{AB}$ indicates the unit vector pointing from $A$ to $B$. Let $H = h_A h_B h_C$, $H' = h_A h_B h_C h_D = H h_D$ and $V := \mathrm{vol}(\sigma_3)$. Without loss of generality(WLOG), set $[A, B, C]$ to be the unique boundary face in $\sigma_3$ with outwards normal $\mathbf{n}_D$.

It can be easily shown that $-h_A \nabla w_A = \mathbf{n}_D$ [14] and the interpolation fields on $[A, B, C]$ only come from itself and its upper-adjacent neighbors. We first check whether its neighbors contribute normal components. WLOG, observe $[C, D, A]$ and we have:

$$\mathbf{w}_{[C,D,A]} \cdot \mathbf{n}_D = w_A (\nabla w_C \times \nabla w_D) \cdot \mathbf{n}_D + w_C (\nabla w_D \times \nabla w_A) \cdot \mathbf{n}_D + w_D (\nabla w_A \times \nabla w_C) \cdot \mathbf{n}_D = 0 \tag{23}$$

The first and second terms vanish since $-h_D \nabla w_D = \mathbf{n}_D$ while the last term vanishes since $w_D = 0$ on face $[A, B, C]$. Therefore, the only normal component contributor is $[A, B, C]$ itself and we have:

$$\int_{[A,B,C]} \mathbf{w}_{[A,B,C]} \cdot \mathbf{n}_D \, dS = 2 \int_{[A,B,C]} \sum_{\mathrm{cyc}} w_A (\nabla w_B \times \nabla w_C) \cdot \mathbf{n}_D \, dS \tag{24}$$

$$= 2 \int_{[A,B,C]} \left( \frac{\cos \theta_A}{h_B h_C} w_A + \frac{\cos \theta_B}{h_C h_A} w_B + \frac{\cos \theta_C}{h_A h_B} w_C \right) dS \tag{25}$$

$$= \frac{2}{3} \mathrm{vol}([A, B, C]) \left( \frac{\cos \theta_A}{h_B h_C} + \frac{\cos \theta_B}{h_C h_A} + \frac{\cos \theta_C}{h_A h_B} \right) \tag{26}$$

$$= \frac{2 \mathrm{vol}([A, B, C])}{3H} \sum_{\mathrm{cyc}} h_A \cos \theta_A \tag{27}$$

in which $\theta_A$ is the intersection angle of $\overrightarrow{DA}$ and $\mathbf{n}_D$, cyc indicates the cyclic summation with respect to the tuple $(A, B, C)$. Thus in the sense of average approximation, the loss on boundary condition $\mathbf{n} \cdot \mathbf{B} = 0$ is

$$\mathcal{L}_{\mathbf{n} \cdot \mathbf{B}} := \frac{4}{9} \sum_{\sigma_2 \in \partial \star \mathcal{M}} \mathbf{h}_{\sigma_2}^2 \left| \frac{\mathrm{vol}(\sigma_2)}{H_{\sigma_2}} \sum_{\mathrm{cyc}} h_A \cos \theta_A \right|^2 \tag{28}$$

**Boundary Condition: $\mathbf{n} \times \mathbf{H} = 0$.** WLOG, assume only $[A, B]$ is on the boundary in $\sigma_3$ and it is obvious that the fields on $[A, B]$ comes from edges $\{\sigma_1' : \exists \sigma_2', \sigma_2, \sigma_1' \prec \sigma_2' \prec \sigma_3 \wedge \sigma_1 \prec \sigma_2 \prec \sigma_3\}$. First, observe that none in this set, except $[A, B]$ itself, has a tangential component contribution. WLOG,

$$\mathbf{w}_{[A,C]} \cdot \overrightarrow{AB} = (w_A \nabla w_C - w_C \nabla w_A) \cdot \overrightarrow{AB} = 0 \tag{29}$$

The first term vanishes since $\nabla w_C \cdot \overrightarrow{AB} = 0$ while the second vanishes since $w_C = 0$ on edge $[A, B]$. And thus the circulation on edge $[A, B]$ is:

$$\int_{[A,B]} \mathbf{w}_{[A,B]} \cdot \overrightarrow{AB} = \int_{[A,B]} (w_A \nabla w_B - w_B \nabla w_A) \cdot \overrightarrow{AB} \tag{30}$$

$$= \frac{\mathrm{vol}([A, B])}{2} (\nabla w_B \cdot \overrightarrow{AB} + \nabla w_A \cdot \overrightarrow{BA}) \tag{31}$$

$$= \frac{\mathrm{vol}([A, B])}{2} \left( \frac{\cos \theta_B}{h_B} + \frac{\cos \theta_C}{h_C} \right) \tag{32}$$

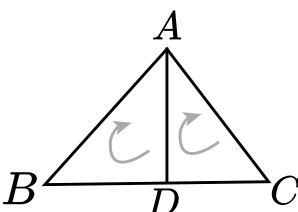

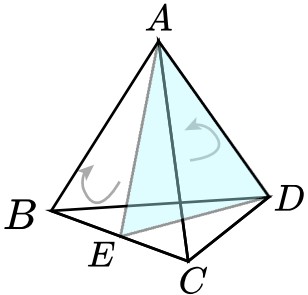

(a) Estimate the integrated dual 1-form on $[A, D]$ via integrated primal 1-forms on $\partial[A, B, C]$.

(b) Estimate the integrated dual 1-form on $[A, E, D]$ via integrated primal 1-forms on $\partial[A, B, C, D]$.

in which $\cos \theta_B = \mathbf{n}_B \cdot \overrightarrow{AB}$. Whence the corresponding loss is:

$$\mathcal{L}_{\mathbf{n} \times \mathbf{H}} := \frac{1}{4} \sum_{\sigma_1 \in \partial \star \mathcal{M}} \mathbf{h}_{\sigma_1}^2 \left| \mathrm{vol}(\sigma_1) \sum_{v \in \partial \sigma_1} \frac{\cos \theta_v}{h_v} \right|^2. \tag{33}$$

**Boundary Condition $\mathbf{n} \times \mathbf{B} = \mathbf{0}$.** This condition is less common than $\mathbf{n} \cdot \mathbf{B} = 0$ in electromagnetism. It involves more terms since all faces of $\sigma_3$ have contributions. The term from $[A, B, C]$ is:

$$\vec{I}_{[A,B,C]} = 2 \int_{[A,B,C]} w_A \frac{\cos \theta_A}{h_B h_C} \overrightarrow{HA} + w_A \frac{\cos \theta_B}{h_C h_A} \overrightarrow{HB} + w_A \frac{\cos \theta_C}{h_A h_B} \overrightarrow{HC} \tag{34}$$

$$= \frac{2 \, \mathrm{vol}([A, B, C])}{3H} \sum_{\mathrm{cyc}} h_A \cos \theta_A \overrightarrow{HA} \tag{35}$$

The terms from $[C, D, A], [A, D, B], [B, D, C]$ are respectively:

$$\vec{I}_{[A,B,C],[C,D,A]} = \frac{V}{h_D H} (h_A \overrightarrow{BA} + h_C \overrightarrow{BC}) \tag{36}$$

$$\vec{I}_{[A,B,C],[A,D,B]} = \frac{V}{h_D H} (h_B \overrightarrow{CB} + h_A \overrightarrow{CA}) \tag{37}$$

$$\vec{I}_{[A,B,C],[B,D,C]} = \frac{V}{h_D H} (h_C \overrightarrow{AC} + h_B \overrightarrow{AB}). \tag{38}$$

Thus, the loss is:

$$\mathcal{L}_{\mathbf{n} \times \mathbf{B}} := \sum_{\sigma_2 \in \partial \star \mathcal{M}} \left| \mathrm{sign}(\sigma_2) \mathbf{h}_{\sigma_2} \vec{I}_{\sigma_2} + \sum_{\sigma_2 \neq \sigma_2' \wedge \sigma_2' \prec \sigma_3} \mathrm{sign}(\sigma_2') \mathbf{h}_{\sigma_2'} \vec{I}_{\sigma_2, \sigma_2'} \right|^2 \tag{39}$$

## C   Estimates among Integrated Forms

For a triangle region $[A, B, C]$ whose diameter is dominated by $\varepsilon$ and a smooth function $f$, then direct expansion gives: $\forall \mathbf{x} \in [A, B, C]$,

$$\int_{[A,B,C]} f(\mathbf{x}) dx = S_{[A,B,C]} f(\xi) = S_{[A,B,C]} f(\mathbf{x}) + S_{[A,B,C]} O(\varepsilon) = S_{[A,B,C]} f(\mathbf{x}) + O(\varepsilon^3) \tag{40}$$

And likewise,

$$\int_{[B,C]} f(\mathbf{x}) dx = |BC| f(\mathbf{x}) + O(\varepsilon^2) \tag{41}$$

Let $|AB| := \mathrm{vol}([A, B])$ and assume that $|BD|/|DC| \gg \varepsilon)$, then

$$\omega(\mathbf{x}) \, \mathrm{vol}([A, B, D]) + O(\varepsilon^3) = \int_{[A,B,D]} d\omega = \int_{[A,B]} \omega + \int_{[B,D]} w + \int_{[D,A]} \omega \tag{42}$$

$$w(\mathbf{x}') \operatorname{vol}([A, C, D]) + O(\varepsilon^3) = \int_{[A,C,D]} dw = \int_{[A,C]} w + \int_{[C,D]} w + \int_{[D,A]} w \qquad (43)$$

Note that $\omega(\mathbf{x}') = \omega(\mathbf{x}) + O(\varepsilon)$ and we have:

$$\frac{\int_{[A,B,D]} d\omega}{\int_{[A,C,D]} d\omega} = \frac{\operatorname{vol}([A, B, D]) + O(\varepsilon^3)}{\operatorname{vol}([A, C, D]) + O(\varepsilon^3)} = \frac{|BD| + O(\varepsilon^2)}{|DC| + O(\varepsilon^2)} = \frac{|BD|}{|DC|} + O(\varepsilon) \qquad (44)$$

and

$$\frac{\int_{[A,B,D]} d\omega}{\int_{[A,C,D]} d\omega} = \frac{\int_{[A,B]} \omega + (|BD|/|BC|) \int_{[B,C]} \omega + \int_{[D,A]} \omega + O(\varepsilon^2)}{\int_{[A,C]} \omega - (|CD|/|BC|) \int_{[B,C]} \omega + \int_{[D,A]} \omega + O(\varepsilon^2)} \qquad (45)$$

Combining Eq. 44 and Eq. 45 and rearranging the terms gives:

$$\int_{[A,D]} \omega = \frac{|BD|}{|BC|} \int_{[A,C]} \omega + \frac{|CD|}{|BC|} \int_{[A,B]} \omega + O(\varepsilon^2) \qquad (46)$$

Likewise, one can mimic the process above and obtain the estimation of dual 2-forms by primal 2-forms:

$$\int_{[A,E,D]} \omega = \frac{|BE|}{|BC|} \int_{[A,C,D]} \omega + \frac{|CE|}{|BC|} \int_{[A,B,D]} \omega + O(\varepsilon^3). \qquad (47)$$

This can be generalized to high-dimensional cases. For brevity, let $I(\sigma) := \int_\sigma \omega$ and $\tau_1 := [v_{i_0}, ..., v_{i_{n-2}}, u, w], \tau_2 := [v_{i_0}, ..., v_{i_{n-2}}, w, v]$ and we have:

$$O(\varepsilon) + \frac{\operatorname{vol}([u, w])}{\operatorname{vol}([w, v])} = \frac{\int_{\tau_1} d\omega}{\int_{\tau_2} d\omega} = \frac{\int_{\partial \tau_1} \omega}{\int_{\partial \tau_2} \omega} \qquad (48)$$

, the right-hand side (RHS) equals:

$$\text{RHS} = \frac{\sum_{j=0}^{n-2} (-1)^j I_j(u, w) + (-1)^{n-1} I([v_{i_{1:n-2}}, w]) + (-1)^n I([v_{i_{1:n-2}}, u])}{\sum_{j=0}^{n-2} (-1)^j I_j(w, v) + (-1)^{n-1} I([v_{i_{1:n-2}}, v]) + (-1)^n I([v_{i_{1:n-2}}, w])} \qquad (49)$$

$$= \frac{\sum_{j=0}^{n-2} (-1)^j \frac{\operatorname{vol}([u,w])}{\operatorname{vol}([w,v])} I_j(u, w) + (-1)^{n-1} I([v_{i_{1:n-2}}, w]) + (-1)^n I([v_{i_{1:n-2}}, u]) + O(\varepsilon^{n-1})}{\sum_{j=0}^{n-2} (-1)^j \frac{\operatorname{vol}([u,w])}{\operatorname{vol}([w,v])} I_j(w, v) + (-1)^{n-1} I([v_{i_{1:n-2}}, v]) + (-1)^n I([v_{i_{1:n-2}}, w]) + O(\varepsilon^{n-1})} \qquad (50)$$

in which $I_j(u, w) = I([v_{i_{1:j-1, j+1:n-2}}, u, w]), I_j(w, v) = I([v_{i_{1:j-1, j+1:n-2}}, w, v])$. This gives:

$$\frac{\operatorname{vol}([u, w])}{\operatorname{vol}([w, v])} = \frac{(-1)^{n-1} I([v_{i_{1:n-2}}, w]) + (-1)^n I([v_{i_{1:n-2}}, u])}{(-1)^{n-1} I([v_{i_{1:n-2}}, v]) + (-1)^n I([v_{i_{1:n-2}}, w])} + O(\varepsilon^{n-1}) \qquad (51)$$

$$= \frac{I([v_{i_{1:n-2}}, w]) - I([v_{i_{1:n-2}}, u])}{I([v_{i_{1:n-2}}, v]) - I([v_{i_{1:n-2}}, w])} + O(\varepsilon^{n-1}) \qquad (52)$$

combine with the volume decomposition identity $\operatorname{vol}([u, w]) + \operatorname{vol}([w, v]) = \operatorname{vol}([u, v])$ and we have:

$$I([v_{i_{1:n-2}}, w]) = \frac{\operatorname{vol}([w, v])}{\operatorname{vol}([u, v])} I([v_{i_{1:n-2}}, u]) + \frac{\operatorname{vol}([u, w])}{\operatorname{vol}([u, v])} I([v_{i_{1:n-2}}, v]) + O(\varepsilon^{n-1}). \qquad (53)$$

### C.1 Estimate Integrated Dual Forms from Integrated Primal Forms

We only propose ways to estimate dual 1-forms in $\mathbb{R}^2$ and $\mathbb{R}^3$ and 2-forms from primal 1-forms and 2-forms since usual integration quadrature can be leveraged directly. Apart from that, we further show how to estimate dual 1-forms from primal 1-forms in $\mathbb{R}^n$. Higher-order estimates and higher dimensions are left for future work.

$O$ is a non-overlappling vertex with vertices $A, B, C$ s.t. $O \in \operatorname{span}[A, B, C]$. Specifically, if $O$ is the dual node and the circumcenter of $[A, B, C]$, then $I([O, \star[B, C]]) = (I([O, B]) + I([O, C]))/2$. Therefore, it is reduced to show the relation between $I([O, X])$ and $I([X, Y])$ where $X, Y \in \{A, B, C\}$.

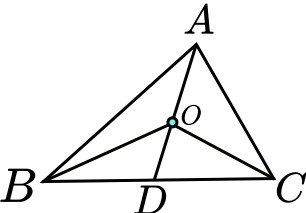

Figure 9: Estimate $\Omega^1(\star\mathcal{M})$ by $\Omega^1(\mathcal{M})$ in $\mathbb{R}^2$

Note that $I([A,O]) = \frac{\text{vol}([A,O])}{\text{vol}([A,D])} I([A,D])$ and $I([A,D])$ can be estimated by $I([A,B])$ and $I([A,C])$. And it appears done. However, it is possible to avoid introducing a new vertex $D$, in which case it is easier to extend to higher-dimensional cases. For brevity, let $S_A := \text{abs}(\text{vol}(B,C,O)), S := \text{vol}([A,B,C])$ and $S_B, S_C$ is likewise defined.

$$\frac{\text{vol}([C,D])}{\text{vol}([B,C])} = \frac{\text{vol}([A,C,D])}{\text{vol}([A,C,B])} = \frac{\text{vol}([O,C,D])}{\text{vol}([O,C,B])} = \frac{S_B}{S_C + S_B} = \frac{S_B}{S - S_A} \tag{54}$$

And thus:

$$\begin{aligned} I([A,O]) &= \frac{\text{vol}([A,O])}{\text{vol}([A,D])} \left( \frac{\text{vol}([C,D])}{\text{vol}([B,C])} I([A,B]) + \frac{\text{vol}([B,D])}{\text{vol}([B,C])} I([A,C]) \right) \\ &= \frac{S - S_A}{S} \left( \frac{S_B}{S - S_A} I([A,B]) + \frac{S_C}{S - S_A} I([A,C]) \right) \\ &= \frac{S_B}{S} I([A,B]) + \frac{S_C}{S} I([A,C]) \end{aligned} \tag{55}$$

Combine with $I([O,\star[B,C]]) = (I([O,B]) + I([O,C]))/2$, we have:

$$I([O,\star[B,C]]) = \frac{S_A}{S} I([A,B]) + \frac{S_B - S_C}{S} I([B,C]) - \frac{S_A}{S} I([C,A]) \tag{56}$$

namely,

$$\begin{pmatrix} I[O,\star[B,C]] \\ I[O,\star[C,A]] \\ I[O,\star[A,B]] \end{pmatrix} = \frac{1}{2S} \begin{pmatrix} S_A & S_B - S_C & -S_A \\ -S_B & S_B & S_C - S_A \\ S_A - S_B & -S_C & S_C \end{pmatrix} \begin{pmatrix} I([A,B]) \\ I([B,C]) \\ I([C,A]) \end{pmatrix} + O(\varepsilon^2) \tag{57}$$

However, this linear transform is not invertible since its determinant is zero. This implies that one cannot recover the integrated primal forms simply by this transform formula. We will later turn to this topic in Appendix C.2.

Fortunately, this formula can be easily extended to dimension $n$ with the volume decomposition identity. For $\sigma_n = [v_0, ..., v_n]$ and $u \in \text{span}\, \sigma_n$, we have:

$$\sum_k (-1)^\tau \text{vol}([\tau(\sigma_{-k}), u]) = \text{vol}(\sigma_n) \tag{58}$$

in which $\sigma_{-k} := \sigma_n - \{v_k\}$ and $\tau$ is the permutation of the rest vertices. This can be easily verified by the identity Eq. 59, in which there are only $n$ non-zero terms within $2^n$ terms in total.

$$\begin{aligned} \text{vol}(\sigma_n) &= \det(v_1 - v_0, v_2 - v_0, \ldots, v_n - v_0) \\ &= \det((v_1 - u) + (u - v_0), \ldots, (v_n - u) + (u - v_0)) \\ &= \sum_k (-1)^\tau \text{vol}([\tau(\sigma_{-k}), u]) \end{aligned} \tag{59}$$

Then we claim:

**Theorem 4.** *For a simplex $\sigma = (u, v, w_1, ..., w_n)$, $\forall o \in \text{int}\, \sigma$,*

$$I([u,o]) = \sum_{w \in \sigma_{-u}} \frac{\text{vol}([o, \sigma_{-w}])}{\text{vol}(\sigma)} I([u,w]) + O(\varepsilon^2). \tag{60}$$

**Proof.** WLOG, consider a simplex $\sigma = (u, v, w_1..., w_m)$, then for $o \notin \text{span}\{w_i\}$, then $\exists! p \in [w_{1:m}, v] \cap [u, o], q \in [w_{1:m}] \cap [v, p]$.

$$I([u, o]) = \frac{\text{vol}([u, o])}{\text{vol}([u, p])} I([u, p]) \tag{61}$$

$$= \frac{\text{vol}([u, o])}{\text{vol}([u, p])} \left( \frac{\text{vol}([p, q])}{\text{vol}([v, q])} I([u, v]) + \frac{\text{vol}([v, p])}{\text{vol}([v, q])} I([u, q]) \right) \tag{62}$$

$$= \frac{\text{vol}([u, o])}{\text{vol}([u, p])} \left( \frac{\text{vol}([p, q])}{\text{vol}([v, q])} I([u, v]) + \frac{\text{vol}([v, p])}{\text{vol}([v, q])} \sum_{w \in \sigma_{-v}} \frac{\text{vol}([\sigma_{-\{v, w\}}, q])}{\text{vol}([\sigma_{-\{v, w\}}, w])} I([u, w]) \right) \tag{63}$$

Whence we only need to check the coefficients in Eq. 60. We will use a property of vol. If $\text{span}\,\sigma = \text{span}\,\tau$ and $\text{vol}\,\sigma = \text{vol}\,\tau$, then for any simplex $\delta$ whose vertex is not in $\text{span}\,\sigma$, we have $\text{vol}([\sigma, \delta]) = \text{vol}([\tau, \delta])$.

The first term is:

$$\frac{\text{vol}([u, o])}{\text{vol}([u, p])} \frac{\text{vol}([p, q])}{\text{vol}([v, q])} = \frac{\text{vol}([u, o, q])}{\text{vol}([u, p, q])} \frac{\text{vol}([u, p, q])}{\text{vol}([u, v, q])} \tag{64}$$

$$= \frac{\text{vol}([u, o, q, \{w_i\}_{-k}])}{\text{vol}([u, v, q, \{w_i\}_{-k}])} \quad , \forall k \tag{65}$$

$$= \frac{\text{vol}([u, o, \{w_i\}])}{\text{vol}([u, v, \{w_i\}])} = \frac{\text{vol}([o, \sigma_{-v}])}{\text{vol}(\sigma)} \tag{66}$$

The second term is:

$$\frac{\text{vol}([u, o])}{\text{vol}([u, p])} \frac{\text{vol}([v, p])}{\text{vol}([v, q])} \frac{\text{vol}([\sigma_{-v-w}, q])}{\text{vol}([\sigma_{-v-w}, w])} = \frac{\text{vol}([u, o])}{\text{vol}([u, p])} \frac{\text{vol}([v, p, \sigma_{-w-v}])}{\text{vol}([v, q, \sigma_{-w-v}])} \frac{\text{vol}([\sigma_{-w}, q])}{\text{vol}([\sigma_{-v-w}, w])} \tag{67}$$

$$= \frac{\text{vol}([u, o])}{\text{vol}([u, p])} \frac{\text{vol}([p, \sigma_{-w}])}{\text{vol}\,\sigma} \tag{68}$$

$$= \frac{\text{vol}([o, \sigma_{-w}])}{\text{vol}\,\sigma} \tag{69}$$

This completes the proof.

By Eq. 60, we obtain all $I([u, o]), u \in \sigma$. But what about $I([\star\sigma_{-u}, o])$? Note that $[\star\sigma_{-u}, o] \subset [o, \sigma_{-u}]$ and thus we can apply Eq. 60 once again in $[o, \sigma_{-u}]$, denoted by $\tau_u$. Consequently, we have:

$$I([o, \star\sigma_{-u}]) = -\sum_{v \in \tau_u} \frac{\text{vol}([(\tau_u)_{-v}, o])}{\text{vol}\,\tau_u} I([v, o]) + o(\varepsilon) \tag{70}$$

$$= -\sum_{v \in \tau_u} \sum_{w \in \sigma_{-v}} \frac{\text{vol}([(\tau_u)_{-v}, o])}{\text{vol}\,\tau_u} \frac{\text{vol}([\sigma_{-w}, o])}{\text{vol}\,\sigma} I([v, w]) + O(\varepsilon^2) \tag{71}$$

Note that for a simplex $\sigma = (x_1, ...x_m, u, v, w_1, ..., w_n)$, there exists and only exists one intersection $p$ between $(u, o)$ and $(v, w_{1:n})$ for all $o \in \text{int}\sigma$ and also only one $q$ intersected at $(v, p)$ and $(w_1, ..., w_m)$. Therefore, combine this result with Eq. 53, repeat the proof above and we have:

**Corollary 5.** *For a simplex* $\sigma = (x_1, ...x_m, u, v, w_1, ..., w_n), \forall o \in \text{int}\sigma,$

$$I([x_{1:m}, u, o]) = \sum_{w \notin \{u, x_{1:m}\}} \frac{\text{vol}([o, \sigma_{-w}])}{\text{vol}(\sigma)} I([x_{1:m}, u, w]) + O(\varepsilon^{m+2}). \tag{72}$$

The estimation of dual 2-forms by primal 2-forms, as a special case, is:

$$I([A, E, D]) = \frac{\text{vol}([B, E])}{\text{vol}([B, C])} I([A, C, D]) + \frac{\text{vol}([C, E])}{\text{vol}([B, C])} I([A, B, D]) \tag{73}$$

$$I([A, E, D]) = \frac{\text{vol}([E, D])}{\text{vol}([D, H])} I([A, H, D]) = \frac{\text{vol}([E, D])}{\text{vol}([D, H])} \frac{\text{vol}([A, H])}{\text{vol}([A, O])} I([A, O, D]) \tag{74}$$

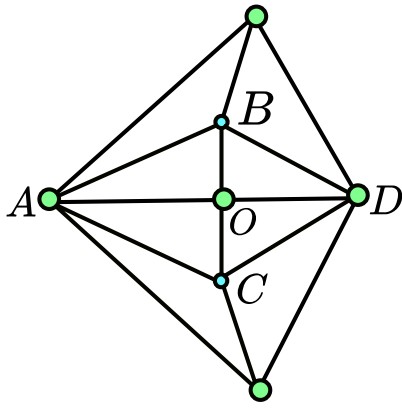

(a) Estimate primal forms by neighbors.

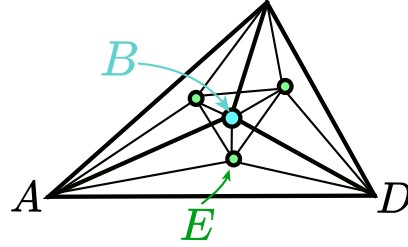

(b) Estimate primal forms by sub-division.

Figure 10: Illustrations for integrated primal form estimation.

$$I([A,O,D]) = \frac{\mathrm{vol}([A,O])}{\mathrm{vol}([A,H])}\frac{\mathrm{vol}([D,H])}{\mathrm{vol}([E,D])}\left(\frac{\mathrm{vol}([B,E])}{\mathrm{vol}([B,C])}I([A,C,D]) + \frac{\mathrm{vol}([C,E])}{\mathrm{vol}([B,C])}I([A,B,D])\right) \tag{75}$$

$$= \frac{\mathrm{vol}([A,O])}{\mathrm{vol}([O,H])}\frac{\mathrm{vol}([A,H])}{\mathrm{vol}([A,O])}\frac{V_B + V_C}{V}\left(\frac{V_C}{V_C + V_B}I([A,B,D]) + \frac{V_B}{V_C + V_B}I([A,C,D])\right) \tag{76}$$

$$= \frac{V_C}{V}I([A,B,D]) + \frac{V_B}{V}I([A,C,D]) \tag{77}$$

## C.2 Estimate Integrated Primal Forms from Integrated Dual Forms

One may ask why we need to estimate the integrated primal forms by dual forms. After all, the dual manifold is not tetrahedralized and thus causes much more complexity in estimation. It stems from the intrinsic requirement in our previous proof. The approximation $\int_\sigma d\omega / \int_\tau d\omega \approx \mathrm{vol}\,\sigma / \mathrm{vol}\,\tau, x \in \sigma$ is fine if $\omega$ is **not exact**, i.e., $d\omega \neq 0$. Any exact forms cannot be properly estimated by the aforementioned quadratures.

But, fortunately, the converse estimation is a remedy to this limitation. Since $\int_{\star\sigma} d^*\omega / \int_{\star\tau} d^*\omega \approx \mathrm{vol}\,\star\sigma / \mathrm{vol}\,\star\tau$ fails if $d^*\omega = 0$. In smooth cases, forms on $\mathcal{M}$ are also forms on $\star\mathcal{M}$. If this also holds in the discrete cases, then by Hodge decomposition, the approximation is valid for any form up to a harmonic form.

Two methods can be applied: **sub-division** and **estimating from neighbors**. The latter has a default: it fails if this dual edge is on the boundary, i.e., it has no neighbors. For brevity, only estimating primal 1-forms from dual 1-forms in $\mathbb{R}^2$ is discussed, which, however, is not challenging to extend to higher-dimensional spaces and forms.

As shown in Figure 10a, given integrated dual forms, integrations of $[A,B], [B,D], [A,C]$, $[C,D], [B,C]$ are all tractable. Then again by Stoke's theorem:

$$\frac{\mathrm{vol}([A,B,D])}{\mathrm{vol}([A,C,D])} \approx \frac{\int_{[A,B,D]} d\omega}{\int_{[A,C,D]} d\omega} = \frac{\int_{[A,B]}\omega + \int_{[B,D]}\omega + \int_{[D,A]}\omega}{\int_{[A,C]}\omega + \int_{[C,D]}\omega + \int_{[D,A]}\omega}, \tag{78}$$

in which $\int_{[A,D]}\omega$ can be estimated. But this approach fails when $[A,D]$ is on the boundary since such a quadrilateral $ABDC$ does not exist. Figure 10b offers another approach. By subdividing the primal triangles, note that $[A,D]$ is contained in both $[A,D,B]$ and $[A,D,E]$ where the integrated forms are known on every edge except $[A,D]$. Therefore, replace $C$ in Eq. 78 by $E$ and we obtain the estimate.

# D  Universal Approximation on Poission Equation

In this section, we show that for any properly exposed Dirichlet boundary value problems, there exists a proposed Higher-order GNN network that approximates the continuous operator. Our proof follows the famous result on the universal approximation property of neural networks for operator.

In our higher-order architecture, we encode the field vectors into differential forms and decode them via Whitney interpolation. Formally,

$$\text{NN} = \mathcal{D} \circ \mathcal{L}_N ... \circ \mathcal{L}_1 \circ \mathcal{E} \tag{79}$$

in which $\mathcal{L}_i, \mathcal{E}, \mathcal{D}$ are the $i$-th layer, encoder layer and decoder layer, respectively. We first show that this encoding-decoding method has a pseudo-inverse property and an asymptotic identity property in a proper function space. First, we consider a simple but heuristic example, a real-valued function $\varphi \in W^{1,2}(I)$ on a closed interval $I \subset \mathbb{R}$. WLOG, $I := [0, 1]$.

**Proof of Theorem 2.**

**Proof.** The existence of $\mathcal{D}_N$ is trivial since for all $c \in \mathbb{R}$, there exists $a \in \mathbb{R}$ such that $\int_{x_i}^{(x_i + x_{i+1})/2} a((x_i + x_{i+1})/2 - x)dx + \int_{(x_i + x_{i+1})/2}^{x_{i+1}} a(x - (x_i + x_{i+1})/2) = c$. Since there are only finite non-smooth points, i.e., the interval endpoints and the midpoints, one can use a smooth mollifier there without changing the integral.

Recall the canonical mollifier $\eta_\varepsilon := \eta(x/\varepsilon)/\varepsilon$ [52] and $\varphi_\varepsilon := \eta_\varepsilon * f$. Since $\varphi \in H^2(I)$ is continuous, $\varphi_\varepsilon \to \varphi$ uniformly on the compact set $I$ in the sense of $L^2$. Thus, we only need to show that $\varphi_\varepsilon \to \tilde{\varphi}$ in $L^2$. Since $\varphi_\varepsilon, \tilde{\varphi} \in C^\infty(I)$, we have $M_1 := \sup_{x \in I} \max\{\varphi'_\varepsilon(x), \tilde{\varphi}'(x)\} < \infty$ and $M_0 := \sup_{x \in I} \max\{\varphi(x), \tilde{\varphi}(x)\}$.

Direct expansion gives:

$$\int_{x_i}^{x_{i+1}} \varphi_\varepsilon(x)dx = \int_{x_i}^{x_{i+1}} \varphi_\varepsilon(s) + \varphi'_\varepsilon(\zeta_s)(x - s)dx = \varphi_\varepsilon(s)2^{-N} + \varphi'_\varepsilon(\zeta_s)\frac{(x_{i+1} - s)^2}{2} \tag{80}$$

Since $\varphi_\varepsilon \to \varphi$ uniformly, there exists $\varepsilon$ small enough such that $\forall \varepsilon_0 > 0$,

$$\frac{\varepsilon_0}{2M_1} + \varphi_\varepsilon(s)2^{-N} + \varphi'_\varepsilon(\zeta_s)\frac{(x_{i+1} - s)^2}{2} = \tilde{\varphi}(t)2^{-N} + \tilde{\varphi}'(\zeta_t)\frac{(t - x_i)^2}{2}$$

which gives:

$$|\varphi_\varepsilon(s) - \tilde{\varphi}(t)| \leq 2^N |\varphi'_\varepsilon(\zeta_s)|\frac{(x_{i+1} - s)^2}{2} + 2^N |\tilde{\varphi}'(\zeta_t)|\frac{(x_{i+1} - t)^2}{2} + \frac{\varepsilon_0}{2M_1} \tag{81}$$

$$\leq 2 \cdot M_1 2^{-N-1} + \frac{\varepsilon_0}{2M_1} \tag{82}$$

$$< \delta + \frac{\varepsilon_0}{2M_1} \tag{83}$$

Thus $\exists N > \log_2 M_1/\delta$ such that $\forall s, t \in [x_i, x_{i+1}], |\varphi_\varepsilon(s) - \tilde{\varphi}(t)| < \delta$. Whence with $\delta = \varepsilon_0/2M_0$

$$\int |\varphi(x) - \tilde{\varphi}(x)|^2 dx \leq \int |\varphi(x) - \varphi_\varepsilon(x)||\varphi(x) - \tilde{\varphi}(x)|dx + \int |\varphi_\varepsilon(x) - \tilde{\varphi}(x)||\varphi(x) - \tilde{\varphi}(x)|dx \tag{84}$$

$$\leq 2M_0(\delta + \frac{\varepsilon_0}{2M_1}) + 2M_0\frac{\varepsilon_0}{2M_0} \tag{85}$$

$$= 3\varepsilon_0 \tag{86}$$

**Corollary 6.** *For uniformly bounded space* $B_M([0,1]) := \{\varphi \in L^2([0,1]) : ||\varphi||_\infty < M\}$, $\forall \varphi \in B_M([0,1])$, *exists* $M$ *such that for all* $N > M$,

$$||\varphi - \mathcal{D}_N \circ \mathcal{E}_N(\varphi)||_{L^2} < \varepsilon.$$

The proof can be analogously extended to higher dimension $\Omega$ with $C^1$ boundary by hypercube-partition.

**Corollary 7.** *Given a simplex-partition of $[0,1]^d$ $\{X_\alpha\}$ with measure-zero intersections, for $\mathbf{v} \in \mathbb{L}^2([0,1]^d) \cap \{\mathbf{u} : ||\mathbf{u}||_\infty < M\}$, $\mathcal{E}_N$ encodes the line integral along every edge of $X_\alpha$ while $\mathcal{D}_N$ preserves them. Then there exists $\delta$ such that for $\sup_\alpha diamX_\alpha < \delta$,*

$$||\mathbf{u} - \mathcal{D}_\delta \circ \mathcal{E}_\delta(\mathbf{u})||_{L^2} < \varepsilon.$$

**Proof.** Consider a hypercube partition where the edges align with the coordinate axes. Therefore $\mathbf{u} \cdot \mathbf{t}_1$ can be viewed a function on $x_1$-axis, which is reduced to Theorem 1 and we have:

$$||\mathbf{u} - \mathcal{D}_N \circ \mathcal{E}_N(\mathbf{u})||_2 \le \sum_{i=1}^d ||(\mathbf{u} - \mathcal{D}_N \circ \mathcal{E}_N(\mathbf{u})) \cdot \mathbf{t}_i||_2 < dM\varepsilon \tag{87}$$

Specifically, in $\mathbb{R}^3$, encoding $\mathbf{u}$ onto faces in the form of flux through the face is also proper since $\mathbf{u} \cdot \mathbf{n}_i$ is equivalent to encoding that to the edges. It is because faces are of codimension 1 in essence.

Define the projection operator $\mathcal{P}_N := \mathcal{D}_N \circ \mathcal{E}_N$. For $\mathcal{D}_N$, the degree of freedom of $\mathcal{D}_N$ is $2^N$, and the interpolation is usually continuous with the encoded integral features in $L^2$-sense. And the integral operator $\mathcal{E}_N$ is continuous since $\mathcal{E}_N$ is linear and bounded, as the integral domain is compact. Therefore, $\mathcal{P}_N$ is continuous with proper interpolation.

Now consider the continuous functional family $C(K)$ on a compact set $K$. Let $V \subset\subset H^2(K) \cap B_M(K)$ in which $B(K) := \{f : K \to \mathbb{R} | |f'|_\infty < M\}$. By continuity, $\mathcal{P}_N V$ is still compact. Consider the space $W_N := V \cup \mathcal{P}_N V$ and $W := \bigcup_{i=0}^\infty W_i = V \cup \bigcup_{i=0}^\infty \mathcal{P}_i V$.

**Lemma 8** (Arzela-Ascoli)**.** *$X$ is a Banach space and $K \subset\subset X$. A subset $V$ of $C(K)$ has compact closure if and only if $V$ is uniformly bounded and equicontinuous.*

**Theorem 9.** *$W$ is closed, uniformly bounded and equicontinous. By Arzela-Ascoli's lemma, $W$ is compact.*

**Proof.** Consider a function sequence $\{f_n\}$. If $\{f_n\}$ has a subsequence in $V$, then it converges to $f$ in $V$ since $V$ is compact. If $\{f_n\}$ has a subsequence $\{f_{i_n}\}$ such that $\sup_n i_n < M$, then it is again completed as the finite union preserves compactness. Therefore, we only need to consider the case $\{\mathcal{P}_{i_n} f_n\}$ in which $f_n \in V, i_n$ increasing to infinity. Note that $\{f_n\}$ must have a subsequence converging to $g$ in $V$ and thus:

$$||\mathcal{P}_{i_n} f_n - g||_2 \le ||\mathcal{P}_{i_n} f_n - f_n||_2 + ||f_n - g||_2 < \delta/2 + \delta/2 < \delta \tag{88}$$

Thus $\lim_{n\to\infty} \mathcal{P}_{i_n} f_n = g$ a.e. $W$ is bounded due to $|f'|_\infty < M$ and Poincaré inequality. $W$ is equicontinuous if the interpolation is Lipschitz. For instance, the linear interpolation has a Lipschitz constant $2\sqrt{C}M$.

$$\left(\int_{x_i}^{x_{i+1}} f(x)dx\right)^2 \le \int_{x_i}^{x_{i+1}} f(x)^2 dx \int_{x_i}^{x_{i+1}} 1^2 dx \tag{89}$$

$$\le C2^{-N} \int_{x_i}^{x_{i+1}} f'(x)^2 dx \tag{90}$$

$$\le CM^2 2^{-2N} \tag{91}$$

Whence functions in $W$ are all Lipschitz and have a common upper bound $2\sqrt{C}M$ and are therefore equicontinuous. By Arzela-Ascoli Lemma, $W$ is compact.

Consider a continuous convolution operator

$$\mathcal{G}[g](x) := \int_\Omega f(x-y)g(y)dy \tag{92}$$

and suppose that $\mathcal{G} : V \to W$, then we have:

$$||\mathcal{G}[g] - \text{NN}[g]||_2 = ||\mathcal{G}[g] - \mathcal{P}_n\mathcal{G}[g] + \mathcal{P}_n\mathcal{G}[g] - \mathcal{D} \circ \mathcal{L} \circ \mathcal{E}[g]||_2 \tag{93}$$

$$\le ||\mathcal{G}[g] - \mathcal{P}_n\mathcal{G}[g]||_2 + ||\mathcal{P}_n\mathcal{G}[g] - \mathcal{D} \circ \mathcal{L} \circ \mathcal{E}[g]||_2 \tag{94}$$

$$\le \sup_{h \in W} ||(I - \mathcal{P}_n)\mathcal{G}[h]||_2 + ||\mathcal{D} \circ (\mathcal{E} \circ \mathcal{G}[g] - \mathcal{L} \circ \mathcal{E}[g])||_2 \tag{95}$$

Note that $W$ is compact and so is $\mathcal{G}W := \{\mathcal{G}[g] : g \in W\}$, therefore

$$\lim_{n \to \infty} \sup_{h \in W} ||(I - \mathcal{P}_n)\mathcal{G}[h]||_2 = 0$$

Note that in the linear interpolation scheme, $||\mathcal{D}(\mathbf{h}_1 - \mathbf{h}_2)||_2 = ||\mathbf{h}_1 - \mathbf{h}_2||_1$. Thus, the only thing to show is that $\mathcal{E} \circ \mathcal{G}[g] - \mathcal{L} \circ \mathcal{E}[g]$ can be bounded with a large enough $N$.

**Theorem 10** (Green Function [52]). *The Green function of Poisson problem in $\mathbb{R}^n$ is:*

$$G(\mathbf{x}, \mathbf{y}) = \begin{cases} \frac{1}{(n-2)\alpha_n} ||\mathbf{x} - \mathbf{y}||_2^{2-n}, & n \geq 3 \\ \frac{1}{2\pi} \ln ||\mathbf{x} - \mathbf{y}||_2, & n = 2 \end{cases} \tag{96}$$

*in which $\alpha_n$ is the surface area of a $n$-dimensional unit sphere and the solution for Dirichlet problem*

$$\Delta u(\mathbf{x}) = f(\mathbf{x}), \mathbf{x} \in \Omega; u(\mathbf{x}) = g(\mathbf{x}) \equiv 0, \mathbf{x} \in \partial\Omega \tag{97}$$

*is given by*

$$u(\mathbf{x}) = \int_\Omega G(\mathbf{x} - \mathbf{y})f(\mathbf{y})dy - \int_{\partial\Omega} \partial_n G(\mathbf{x} - \mathbf{y})g(\mathbf{y})dy = \int_\Omega G(\mathbf{x} - \mathbf{y})f(\mathbf{y})dy \tag{98}$$

**Lemma 11** (Extension Theorem [52]). *For a bounded closed $\Omega$ with $C^1$-boundary, there exists an extension operator $E : W^{1,p}(\Omega) \to W^{1,p}(\mathbb{R}^n)$, such that $E(u) = u$ in $\Omega$ and $E(u)$ is compactly supported and continuous:*

$$||Eu||_{W^{k,p}(\Omega)} \leq c_{n,\Omega}||u||_{W^{k,p}(\Omega)} \tag{99}$$

For a bounded closed $\Omega \subset \mathbb{R}^n$, it must be paracompact and thus has a smooth partition of unity $\{\rho_\alpha\}$ [53]. For a square-like uniform mesh, one can construct a finite open cover $\{O_\alpha\}$ such that $\forall \delta > 0$, there exists a finite open cover $\sup_{\alpha \neq \beta} \text{diam}(O_\alpha \cap O_\beta) < \delta$ and $O_\alpha$ contains the partitioned hypercube $\alpha$. For those hypercubes not intersected with the boundary, the corresponding open covers can also have no intersections.

Recall that the extension theorem can be shown by the reflection method, in which we can locally extend a $u|_{O_i \cap \Omega} \equiv \rho_i u, \partial\Omega \subset \bigcup_{i=1}^K O_i$ due to the boundary compactness. It is obvious that

$$\mu(\text{supp}(Eu)) \geq \mu(\text{supp}(u)), \tag{100}$$

in which $\mu$ is the Lebesgue measure. And also note that

$$\mu(\text{supp}(Eu)) \leq \mu\left(\text{supp}(u) \cup \text{supp}\left(\sum_{i=1}^K \mathcal{R}_i[\rho_i u]\right)\right) \tag{101}$$

$$\leq \mu(\text{supp}(u)) + \sum_{i=1}^K \mu\left(\text{supp}(\mathcal{R}_i[\rho_i u])\right) \tag{102}$$

$$= \mu(\text{supp}(u)) + \sum_{i=1}^K \mu\left(\text{supp}(\rho_i u)\right) \tag{103}$$

$$\leq \mu(\text{supp}(u)) + \sum_{i=1}^K \mu\left(\text{supp}(\rho_i)\right) \tag{104}$$

$$\leq \mu(\text{supp}(u)) + \delta_0 \tag{105}$$

in which $\mathcal{R}_i$ is the reflection operator concerning the open set $O_i$, and the last holds because ass the partition gets finer, the total volume of boundary hypercubes tends to be zero. Therefore, $f$ defined on $\Omega$ can be extended to $\mathbb{R}^n$ such that $\int_\Omega G(\mathbf{y})E[f](\mathbf{x} - \mathbf{y})dy = \int_{\mathbb{R}^n} G(\mathbf{y})E[f](\mathbf{x} - \mathbf{y})dy + \varepsilon$. For brevity, we omit $\varepsilon$ and identify $E[f]$ as $f$ in the following discussion. Consider a component of

$\mathcal{E} \circ \mathcal{G}[g]$, i.e., the integral on $I$, and we have:

$$\int_I \int_\Omega G(\mathbf{x} - \mathbf{y})f(\mathbf{y})dydx = \int_I \int_\Omega G(\mathbf{y})f(\mathbf{x} - \mathbf{y})dydx \tag{106}$$

$$= \sum_J \int_I \int_J G(\mathbf{y})f(\mathbf{x} - \mathbf{y})dydx \tag{107}$$

$$= \sum_J \int_I \int_J G(\mathbf{y})f(\mathbf{x} - \mathbf{y}_J) - G(\mathbf{y})\nabla f(\mathbf{x} - \zeta_J) \cdot (\mathbf{y} - \mathbf{y}_J)dydx \tag{108}$$

$$= \sum_J \int_I f(\mathbf{x} - \mathbf{y}_J)dx \int_J G(\mathbf{y})dy - \int_I \int_J G(\mathbf{y})\nabla f(\mathbf{x} - \zeta_J) \cdot (\mathbf{y} - \mathbf{y}_J)dydx \tag{109}$$

$$= \sum_J \int_{I+\mathbf{y}_J} f(\mathbf{x})dx \int_J G(\mathbf{y})dy - \int_I \int_J G(\mathbf{y})\nabla f(\mathbf{x} - \zeta_J) \cdot (\mathbf{y} - \mathbf{y}_J)dydx \tag{110}$$

$$=: \sum_J T_{1,I,J} + T_{2,I,J} \tag{111}$$

Note that $\int_I f(y)dy$ are components of $\mathcal{E}[g]$ and thus the neural network $\mathcal{L}$ is required to learn $N^2$ coefficients in the form of $\int_I f(x - y_J)dx$, a continuous function of $x$. But here the integral is on the translated $I$, i.e., $I + \mathbf{y}_J$ in the sense of Minkowski sum. Since $f$ is extended to $\mathbb{R}^n$ such that $f = 0$ a.e. on $\mathbb{R}^n - \Omega$, we can omit those $I + \mathbf{y}_J$ not in the mesh. Therefore, encoding $f(y)$ as $N^n$ integrals is still enough to estimate $T_{1,I,J}$.

For $T_{2,I,J}$, in sub-regions with $f(\mathbf{x})$ is not constant 0, as $N \to \infty$, we must have $|f(\mathbf{x})| > \delta$ due to continuity. Then we have

$$|T_{2,I,J}| \leq \int_J |G(\mathbf{y})| \cdot ||\nabla f(\mathbf{x} - \zeta_J)||_2 \cdot ||\mathbf{y} - \mathbf{y}_J||dy \leq M \operatorname{diam}(J)\left|\int_J G(\mathbf{y})dy\right| \tag{112}$$

Thus as $N \to \infty$, $\operatorname{diam}(J) \to 0 \Rightarrow T_{2,I,J} = o(\int_J G(\mathbf{y})dy)$ while $T_{1,I,J} = O(\int_J G(\mathbf{y})dy)$. We conclude that $T_{2,I,J} = o(T_{1,I,J})$ and therefore, we only need to focus on the main part. The first term can be interpreted as a matrix multiplication where $\mathcal{L}$ receives $x \in \Omega$ as the input and outputs $(N^n)^2$ coefficients. This can be approximated by the canonical universal approximation theorem [54]. This seemingly holds for special meshes, but in fact, one can adopt a bounded Jacobian to deform it into a more generalized mesh.

Note that the operator norm of $\mathcal{G}[f] := \int_\Omega G(\mathbf{x} - \mathbf{y})f(\mathbf{y})dy$ is finite as long as $\Omega$ is closed and bounded. Then by **Proper Encoder-Decoder** theorem, we reach Theorem 3. Following an analogous proof, one can see that it also holds for non-homogeneous Dirichlet conditions. Thus, we have the following corollary:

**Corollary 12.** *For non-homogeneous Dirichlet problems, $\Delta u(\mathbf{x}) = f(\mathbf{x}), \mathbf{x} \in \Omega; u(\mathbf{x}) = g(\mathbf{x}), \mathbf{x} \in \partial\Omega$, there also exists a network universally approximates $u$. Take gradients on both sides, there exists a network that universally approximates $\nabla u$, i.e., the electric intensity $\mathbf{E}$.*

## E    Dataset Details

Table 6 lists out details of several benchmarks. 2D meshes are obtained by adding random points into a 2D convex hull and then triangulated via the open Python library *Triangle* [55]. 3D meshes come from example meshes provided by a free C++ tetrahedralization library *TetGen* [56].

**2D cases**. The 2D-electrostatics case involves triangle pieces constituting a square, including two media: silicon and a non-isotropic linear material. Each piece is assigned a random charge density excitation, and the region boundary condition is set as equal voltage. The 2D-magnetostatics case includes mu-metal (a material with a non-linear B-H curve) triangle pieces, composing an irregular mesh with two holes. Each triangle is randomly assigned a current density. The region boundary condition is *balloon*, implying an infinitely large region.

Table 6: Various meshes are adopted in different BVPs. ELE and MAG are for electrostatics and magnetostatics, respectively. NA means not applicable.

| Mesh | Square | Holes | Socket | Gear |
|---|---|---|---|---|
| Vertex amount | 105 | 180 | 2,407 | 2,453 |
| Edge amount | 277 | 487 | 12,865 | 12,833 |
| Face amount | 173 | 306 | 19,032 | 18,843 |
| Cell amount | NA | NA | 8,580 | 8,462 |
| Train set size | 120 | 120 | 240 | 240 |
| BVP type | 2D-ELE | 2D-MAG | 3D-ELE | 3D-MAG |

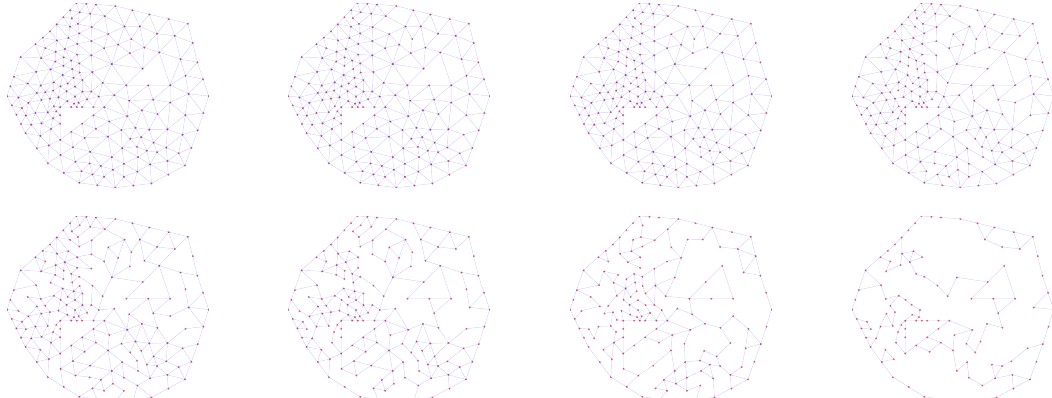

Figure 11: Illustrations of degraded meshes. The edge amount decreases from left to right and then from top to bottom.

| Dropped Edge Amount | 0 | 25 | 50 | 100 | 150 | 200 | 250 | 300 |
|---|---|---|---|---|---|---|---|---|
| Vertex amount | 180 | 180 | 180 | 180 | 177 | 168 | 155 | 114 |
| Edge amount | 487 | 462 | 437 | 387 | 334 | 275 | 216 | 139 |
| Face amount | 306 | 281 | 256 | 206 | 156 | 106 | 60 | 24 |
| AR | 1.286 | 1.301 | 1.318 | 1.365 | 1.401 | 1.458 | 1.595 | 1.929 |
| EAS | 0.595 | 0.594 | 0.607 | 0.670 | 0.783 | 0.811 | 0.975 | 0.816 |
| ATR | 0.197 | 0.251 | 0.316 | 0.404 | 0.487 | 0.584 | 0.657 | 0.860 |

Table 7: Details about the generated degraded meshes.

**3D cases**. The 3D-electrostatics case is an $Al_2O_3$-based socket, in which four $SiO_2$ cubic blocks are embedded with uniformly distributed random charge density. The solving region boundary condition is set to a uniform electric potential to simulate an enclosed empty cell in a metal. The 3D-magnetostatics case is a cobalt quarter-gear with three copper circuits with uniformly distributed random currents. The region boundary condition is that $\mathbf{H}$ is tangential to the boundary and the integral on the boundary vanishes.

All coordinates are normalized to $[0, 1]^d, d = 2, 3$, scalar fields are standardized, and vector fields are divided by average norm to eliminate scaling differences during pre-processing.

**Degenerated Meshes via Dropping Edges**. By randomly dropping a certain amount of edges and further deletion to fix the topology, these degraded meshes illustrated in Figure 11 and depicted in Table 6 are obtained. Usually, a mesh is considered having good quality if its elements are uniform and regular. We briefly introduce these three indicators here:

- **Aspect Ratio (AR)**. There are many versions of AR. We define it as the ratio between the maximum edge length over the minimum.

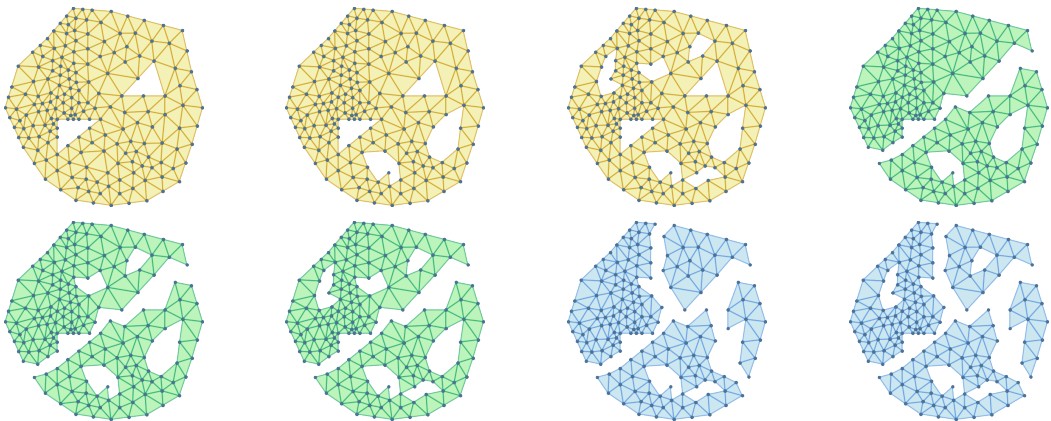

Figure 12: Illustrations of meshes with different topological characteristics. It is clear that those colored identically have the same connect compone

- **Equi-Angle Skewness (EAS)**. An N-polygon's ideal angle is $\theta = \pi(N-2)/N$. For an actual angle $\theta_i$, its offset is $\delta_i := |\theta_i - \theta| / \max\{\theta, \pi - \theta\}$. And its EAS is $\max_i \delta_i$.
- **Area Transition Ratio (ATR)**. It depicts the uniformity of element size. Let $A_i, A_j$ be the area of two neighboring elements. Then its ATR is $|A_i - A_j| / \max\{A_i, A_j\}$.

# F    Implementation Details

The hardware configuration consists of an Ubuntu 20.04 LTS operating system platform powered by four NVIDIA RTX A6000 Graphics Processing Units with 48GB memory. We use the PyTroch framework version 1.13 alongside Python 3.12 interpreter. Model optimization involved selection of the initial learning rate parameter $\gamma$ for the Adam optimizer, which was systematically chosen from the discrete value set $\{i \times 10^{-j} : i \in \{1, 5\}, j \in \{1, 2, 3, 4, 5\}\}$ to achieve optimal performance metrics. Datasets are partitioned into training, validation, and test sets with the ratio 8:1:1. Each model in the main result is trained until the validation loss converges within 1,000 epochs.

