# OpenReview forum: "Boundary-Value PDEs Meet Higher-Order Differential Topology-aware GNNs"
_NeurIPS.cc/2025/Conference — NeurIPS 2025 spotlight_

### Official Review · Reviewer_RTKj · 2025-06-16

**Clarity:** 3
**Significance:** 4
**Originality:** 4
**Rating:** 5
**Confidence:** 4

**Summary:**

This paper introduces a novel method called DEC-HOGNN in the category of neural operators. It makes use of higher-order graph neural networks that can handle interactions between simplices of different orders, not just same orders. It also designs an encoder-decoder structure for differential forms of different orders involved to make predictions. By writing down fields as differential forms, the physical losses can also be easily implemented. This approach is tested on boundary value problems in electrostatics and magnetostatics for both 2D and 3D.

**Questions:**

1. How are the boundary conditions $\nabla \phi = u(\bf{x}), \bf{x} \in \partial \Omega$ implemented as an input for this neural operator?
2. Related to point 1 in weaknesses, why are the data still generated with traditional FEM, instead of FEEC? If data are generated with FEEC, then no encoding is required and all differential forms are naturally solved out.

**Ethical Concerns:**

["NO or VERY MINOR ethics concerns only"]

**Final Justification:**

All of my concerns have been resolved from this rebuttal period. This paper is of good quality, so I do not have further concerns. Therefore, I recommend acceptance for this paper.

**Limitations:**

Yes, the limitations are discussed in this paper.

**Paper Formatting Concerns:**

No obvious concerns are noticed by the reviewer.

**Quality:**

4

**Strengths And Weaknesses:**

Strengths:
1. This is an interesting attempt to implement physical losses elegantly and effectively in neural operators, which an inspiring and meaningful first exploration for AI for PDE community. This method does not require differentiation for physical losses.
2. The mathematical motivations of this work are demonstrated in an easy to follow manner.
3. There are clear proofs for important theorems mentioned in this paper, including one universal approximation property for this proposed neural operator.

Weaknesses:
1. Though a large part of this paper is devoted to FEEC, the data generation is still achieved with traditional FEM methods, but not FEEC. This point makes this work not in a "pure" FEEC manner.
2. The datasets explored in the experiments of this work do not cover a wide enough range of problems. Only electrostatics and magnetostatics are covered, which does not provide enough evidence for the superiority of this approach.
3. The inference speed is not discussed and compared with other methods.

---

> ### Author Rebuttal · Authors · 2025-07-29
>
> We sincerely appreciate your insightful feedback and valuable assessment of our work. Below we address each comment in detail.
>
> > **Q1**. The inference speed is not discussed and compared with other methods.
>
> Thanks for your comment. We have updated the manuscript to compare our model with baselines on computation cost. Below we provide various relative indices on the 2D electrostatics benchmark and 3D electrostatics benchmark respectively.
>
> Since our experiments mainly focused on the effectiveness of the model, there is still much room to reduce the overheads of the model. Also, higher-order elements come with extra expenditures. The adjacency amount soars up when lots of tetrahedrals appear in a finer tetrahedralization, among which some might be unnecessary. An interesting direction for future research is how to fully leverage higher-order topological adjacencies without being hindered by the proliferation of small tetrahedrals. Nevertheless, our results show that the inference efficiency of the proposed model is acceptable and remains significantly faster than classical solvers, which typically require several seconds per inference.
>
> Tab 1: Comparison on the 2D electrostatics benchmark.
>
> | Model            | FLOP(M) | Memory(MB) | Inference Time(ms) |
> | ---------------- | ------- | ---------- | ------------------ |
> | DeepONet         | 0.10    | 2.74       | 0.33               |
> | MKGN             | 0.22    | 19.80      | 3.73               |
> | Galerkin-Type    | 2.96    | 18.20      | 1.67               |
> | GNOT             | 1.99    | 25.30      | 1.55               |
> | Transolver       | 1.34    | 19.05      | 1.84               |
> | GAT-based        | 0.10    | 7.69       | 1.48               |
> | Graph UNet-based | 0.14    | 3.36       | 4.69               |
> | GT-based         | 10.35   | 53.96      | 1.54               |
> | DEC-HOGNN(Ours)  | 2.85    | 55.59      | 5.00               |
>
> Tab 2: Comparison on the 3D electrostatics benchmark.
>
> | Model            | FLOP(M) | Memory(MB) | Inference Time(ms) |
> | ---------------- | ------- | ---------- | ------------------ |
> | DeepONet         | 0.10    | 93.41      | 0.28               |
> | MKGN             | 0.53    | 957.25     | 5.50               |
> | Galerkin-Type    | 0.95    | 1122.03    | 5.29               |
> | GNOT             | 1.99    | 281.99     | 2.13               |
> | Transolver       | 1.35    | 235.51     | 2.46               |
> | GAT-based        | 0.11    | 264.42     | 2.08               |
> | Graph UNet-based | 0.78    | 257.22     | 6.76               |
> | GT-based         | 10.35   | 367.42     | 3.41               |
> | DEC-HOGNN(Ours)  | 5.37    | 9380.33    | 94.09              |
>
> > **Q2**.  How are the boundary conditions $\nabla \phi=u(x)$ implemented as an input for this neural operator?
>
> Usually a scalar potential $\phi$ is unavailable in many physics scenarios. In electromagnetism, $\nabla \phi$ may be represented by electric intensity $\mathbf{E}$ or magnetic intensity $\mathbf{H}$, which can be easily measured by sensors in practice. Therefore, in our approach, we take these vector fields as both input and final output. In fact, we introduce $\phi$ only to show the theorems.
>
> But still, given a scalar field $\phi$, we can leverage the gradient operator defined in graph calculus [1] or simply learn one like NIsoGCN [2] to yield a vector field $\nabla \phi$. It can then be interpreted as a vector proxy of a 1-form and thereby represented as integrated forms on boundary edges.
>
> Ref.
>
> [1] Calculus on Graphs. Joel Friedman et al.
>
> [2] Physics-embedded neural networks: Graph neural pde solvers with mixed boundary conditions.
>
> > **Q3**. Though a large part of this paper is devoted to FEEC, the data generation is still achieved with traditional FEM methods, but not FEEC. This point makes this work not in a "pure" FEEC manner. Related to point 1 in weaknesses, why are the data still generated with traditional FEM, instead of FEEC? If data are generated with FEEC, then no encoding is required and all differential forms are naturally solved out.
>
> Reviewer dQYA has similar concerns. We can fully understand your concerns on the compatibility between traditional FEM and its FEEC and DEC counterparts. For data generation, any of these methods is acceptable, as they yield the same solution for a well-posed PDE. Therefore, the method used for data generation does not compromise the purity or integrity of this work.
>
> Regarding your second point, if input data is presented in a FEEC manner, then we can indeed cast away the encoding process. However, such an assumption can be too ideal to fit more applications. Because it is easy to deploy sensors to sample vector intensity like $\mathbf{B},\mathbf{E}$ while it is hard to measure a differential form down there. Hence, the encoding process is still necessary for a broader applicability.
>
> > **Q4**. The datasets explored in the experiments of this work do not cover a wide enough range of problems. Only electrostatics and magnetostatics are covered, which does not provide enough evidence for the superiority of this approach.
>
> We use Maxwell's equations as a case study, as they encompass a broad range of boundary value problems (BVPs) and exhibit a well-structured formulation within the framework of exterior calculus. Specifically, they involve differential operators such as $\text{div}$ and $\text{curl}$, which also appear in BVPs like *Darcy Flow*, as well as in time-dependent PDEs such as the *Heat and Wave equations*. Upon closer examination, these equations share structural similarities with Maxwell's equations, suggesting they follow the same underlying patterns. Consequently, experiments on electrostatics and magnetostatics may provide evidence for the framework’s broader applicability to a variety of linear PDEs. We will include them in the next version of our paper.
>
> We also mention that this framework can also potentially be applied to non-linear PDEs like *Navier-Stokes Equations (NSE)*. This is mainly because $\text{div},\text{curl}$ can be encoded as higher-order element features. We briefly sketch a possible approach and kindly invite you to the response to **Reviewer 5x2r's Q4**.

---

> > ### Comment · Reviewer_RTKj · 2025-08-04
> >
> > I would like to thank the authors for their answers to my questions. They have resolved my concerns. Since I originally recommend for acceptance for this paper, I will just directly finalize my original rating.

---

### Official Review · Reviewer_dQYA · 2025-07-01

**Clarity:** 4
**Significance:** 3
**Originality:** 4
**Rating:** 5
**Confidence:** 4

**Summary:**

This paper proposed a novel GNN neural operator (NO) that is topology-aware by leveraging discrete exterior calculus (DEC). The advantage of this novel NO includes: 1. preserving the topological structure of the manifold which is also the problem domain of PDE; 2. It can turn some equations involving divergence and curl into boundary conditions by Stokes' theorem, such that the physical loss from equation no longer needs taking derivatives, e.g., divergence and curl, like PINN does.

In order to leverage the DEC, the authors designed a GNN that handles 4 types of neighborhood (see eq (8)-(12)). In experiments of 2D and 3D problems, the proposed method outperforms all baselines and showing strong advantage.

**Questions:**

N.A.

**Ethical Concerns:**

["NO or VERY MINOR ethics concerns only"]

**Final Justification:**

After reading other reviews and rebuttals, I see that all reviews are positive and concerns are addressed. Therefore, my score remains.

**Limitations:**

N.A.

**Paper Formatting Concerns:**

N.A.

**Quality:**

4

**Strengths And Weaknesses:**

**Strength**

This paper is highly innovative and origin by introducing DEC to learning operators with GNN. As mentioned in summary, there are two advantages: 1. Preserving topology structure of the manifold; 2. Simplifying the equation constraints to boundary constraints (see eq (19)-(22)). In prior works, topology structure is either ignored (e.g., transformer-based NO and FNO) or partially encoded (with graph adjacency of GNN). In contrast, DEC utilizes 4 types of neighborhood, which is a complete methodology. Ablation study (Table 2) shows that the complete setting outperforms incomplete ones. The second advantage naturally enables the proposed method to incorporate physics-informed loss without taking derivatives (for some equations, e.g., Maxwell, Navier-Stokes, etc.).

The paper is well written and elegantly presented. Also, for self-completeness, the paper provides universal approximation analysis.

**Weakness**

1. In Table 3, I think the bold numbers are the best of each column. Please clarify accordingly to avoid confusion.
2. The current data is generated with finite element method, which may not be compatible with DEC (as the discussion in line 104-105). If data generation by DEC is not so convenient, this may limit the application of the method to some extent.
3. Typo? On line 476, $H\in\Omega^1(\star M)$ -> $ H\in\Omega^1(M)$ to align with Fig. 2.

---

> ### Author Rebuttal · Authors · 2025-07-29
>
> We sincerely appreciate your constructive feedback and meticulous evaluation of our work. Below, we provide responses to each point raised.
>
> > **Q1**. The current data is generated with finite element method, which may not be compatible with DEC (as the discussion in line 104-105). If data generation by DEC is not so convenient, this may limit the application of the method to some extent.
>
> Thanks for your comments. Reviewer RTKj has similar concerns. We can fully understand your concerns on incompatibility between data and model design. However, FEM-based solvers are significantly more mature and widely adopted than their DEC and FEEC counterparts. Theoretically, all of these solvers can **converge to the same solution**, provided that the PDE is well-posed. As such, FEM, FEEC, and DEC are all valid choices for data generation when no numerical issues arise. This is why data generation using DEC is not strictly necessary.
>
> > **Q2**. *On minor issues*.
> > (i) Typo? On line 476,  $H\in \Omega^1(\star M)\to H\in\Omega^1(M)$  to align with Fig. 2.
> > (ii) In Table 3, I think the bold numbers are the best of each column. Please clarify accordingly to avoid confusion.
>
> Thanks for your careful reading and corrections. For (i),  $H$ should be indeed defined on $M$ since $B$ is on $\star M$; for (ii), we will clarify its meaning in the caption.

---

> > ### Comment · Reviewer_dQYA · 2025-08-06
> >
> > Thank you for the clarification.

---

### Official Review · Reviewer_1rUG · 2025-07-03

**Clarity:** 2
**Significance:** 4
**Originality:** 4
**Rating:** 5
**Confidence:** 3

**Summary:**

This paper proposes a neural operator framework for solving boundary value problems (BVPs) governed by partial differential equations (PDEs). The key innovation lies in leveraging Discrete Exterior Calculus (DEC) and Finite Element Exterior Calculus (FEEC) to explicitly encode higher-order topological structures (nodes, edges, faces, cells) as k-simplex features in graph neural networks (HOGNNs). This allows for principled modeling of scalar and vector fields as differential forms, enabling a topologically-aware neural solver.

The approach includes physics-informed encoder-decoder mappings between vector fields and forms, integrated physical loss functions that correspond to conservation laws (e.g., Gauss’s law), and theoretical guarantees of universal approximation for certain BVP classes. There are limited experimental evaluations on 2D and 3D electrostatics and magnetostatics.

**Questions:**

- The formulation of the neural operator in Eq. (5) is too abstract and hard to interpret until the reader reaches the concrete instantiation in Eq. (7). The authors could either move the motivating example earlier or introduce a simplified case (e.g., scalar Poisson) first.

- Figures 1 and 2 are critical but lack self-contained descriptions. For a technical audience unfamiliar with DEC or the specific PDE being solved, these need extensive legends explaining what each symbol and arrow denotes.

- More focussed experiments: Pick 2-3 shapes in 2D and 3D (exhibiting different topological characteristics), and comprehensively evaluate performance against other state of art. Additionally, while the paper briefly notes that performance is affected by mesh shape and quality, this is an important limitation for deployment in practical engineering settings. Additional empirical analysis (e.g., mesh convergence study) would strengthen the impact.

**Ethical Concerns:**

["NO or VERY MINOR ethics concerns only"]

**Final Justification:**

Thank you to the authors for actively engaging during this process.
I also appreciate the additional results that the authors added.
I am happy to increase my score to 5

**Limitations:**

Yes

**Quality:**

3

**Strengths And Weaknesses:**

*Strengths*

- The methodology is technically sound and grounded in well-established mathematical frameworks (DEC and FEEC), with rigorous theoretical support (e.g., encoder-decoder preservation, universal approximation).
- This work is an important step toward building structure-preserving, physically grounded neural solvers for BVPs, providing a new approach. The proposal to use higher-order GNNs aligned with exterior calculus opens a new direction for learning-based PDE solvers in domains like electromagnetism, fluid dynamics, and elasticity.
- The synthesis of DEC/FEEC with HOGNNs for BVPs appears novel and nontrivial. While GNNs and physics-informed learning are well-trodden, the topological interpretation via differential forms and the dual-manifold message passing are particularly unique contributions (for solving PDEs).
- The physics-informed loss design is very elegant and potentially efficient! I love how the work leverages integrated features (which appear naturally in DEC) directly, which avoids the need for expensive quadrature.

*Weakness*
- This is a very dense paper, with some parts written with excessive jargon (particularly early on), making it hard for a broader audience to parse. Key ideas (like Eq. 5) are underexplained at first reading.
- Figures (e.g., Figure 1 and 2) are central to the understanding but require much more extensive captions and annotations to be fully self-contained.
- It is unclear why the 3D magnetostatic results are not competitive (relative to other baselines) even though the model is designed to generalize naturally to 3D.
-  The comparison misses modern BVP-specific baselines such as FNO, CNO, SCOT, etc., which are more appropriate than DeepONet or Transolver (designed for time-evolving systems).
- The experimental results are very perfunctory, and should be strengthened.
- While the stated focus is on time-dependant PDEs (in the abstract), all analysis and results are for the time-**independant** PDE. This is rather confusing.

---

> ### Author Rebuttal · Authors · 2025-07-29
>
> We are sincerely grateful for the time and effort you have dedicated to reviewing our manuscript. Below, we address each of your comments in detail.
>
> > **Q1**. While the stated focus is on **time-dependent PDEs (in the abstract)**, all analysis and results are for the time-independent PDE. This is rather confusing.
>
> We actually describe **in the abstract** that the **time-independent** boundary value problems (BVPs) in electromagnetism are instantiated to illustrate the proposed framework, not time-dependent.
>
> Though this framework can be easily generalized to time-evolving PDEs, we evaluate our method on time-independent cases, aiming to verify its capability of capturing local behaviors by introducing higher-order elements.
>
> > **Q2**. More focused experiments: Pick 2-3 shapes in 2D and 3D (exhibiting different topological characteristics), and comprehensively evaluate performance against other state of art.
>
> Due to character limit, only the results on 2D magnetotastics are provided. 8 relative benchmarks (from A1 to C2) are used for evaluation, each having different topological properties on connected component amount and hole amount:
>
> |Benchmark Id|A1|A2|A3|B1|B2|B3|C1|C2|
> |-|-|-|-|-|-|-|-|-|
> |Connected Component Amount|1|1|1|2|2|2|4|4|
> |Hole Amount|2|4|8|2|4|8|2|4|
>
> Performance on these benchmarks are compared with other baselines. The table below shows the test loss of each model. It turns out that the advantage of our approach persists as the underlying topology changes.
>
> | Model/Benchmark Id | A1                  | A2                  | A3                  | B1                  | B2                  | B3                  | C1                  | C2                  |
> | ------------------ | ------------------- | ------------------- | ------------------- | ------------------- | ------------------- | ------------------- | ------------------- | ------------------- |
> |FourierType|3.969|3.670|3.532|4.963|4.639|5.561|3.457|4.211|
> |GalerkinType|2.457|2.883|3.207|**1.907**|2.511|2.912|**2.093**|3.275|
> |MKGN|9.177|5.208|6.769|8.428|5.813|4.689|4.259|6.722|
> |Transolver|2.733|$\underline{2.479}$|$\underline{2.167}$|2.301|$\underline{2.273}$|$\underline{2.112}$|2.234|**1.901**|
> |Geo-FNO|9.423|9.448|9.453|9.448|9.450|9.451|9.448|9.451|
> |GNOT|$\underline{2.406}$|3.277|3.009|2.799|3.006|2.588|3.373|2.213|
> |DEC-HOGNN(Ours)|**1.573**|**1.487**|**1.498**|$\underline{2.088}$|**1.901**|**1.727**|$\underline{2.181}$|$\underline{2.176}$|
>
> **Rmk**. Baselines achieving top 1 and top 2 performance are marked via bolding and underscoring, respectively.
>
> > **Q3**. Additionally, while the paper briefly notes that performance is affected by mesh shape and quality, this is an important limitation for deployment in practical engineering settings. Additional empirical analysis (e.g., mesh convergence study) would strengthen the impact.
>
> Thanks for your suggestion. We then analyze the negative impact out of mesh degradation by randomly dropping a certain amount of edges on a mesh hierarchically. Several extra edges are further removed to fix the topology. Due to character limit, only partial data can be demonstrated.
>
> To measure the mesh quality quantitatively, three indicators are adopted where the arrows imply the direction in which the mesh degrades. It is reflected that dropping edges from a triangularized mesh usually comes with mesh degeneration.
>
> Then, we train models on the meshes suffering from different levels of degeneration. The average mesh quality and validation loss within 500 epochs are reported. It is found that minor degeneration would not affect the model performance violently while a major one leads to salient performance drop. Note that it is also infeasible to adopt classical solvers using these meshes with prominent quality issues. Thus these negative effects are tolerable.
>
> | Edge Drop | #E | #V| #F | AR$↑$ | EAS $↑$ | ATR$↑$ | 0     | 100  | 200  | 300  | 400  | 500  |
> |-|-|-|-|-|-|-|-|-|-|-|-|-|
> |0|487|180|306|1.286|0.595|0.197|9.75|4.60|2.95|2.14|1.85|1.44|
> |25|462|180|281|1.301|0.594|0.251|9.32|4.24|2.89|2.18|1.65|1.42|
> |50|437|180|256|1.318|0.607|0.316|9.32|4.44|3.26|2.42|2.01|1.51|
> |100|387|180|206|1.365|0.670|0.404|9.33|4.58|3.52|2.66|2.32|2.04|
> |150|334|177|156|1.401|0.783|0.487|9.41|4.90|3.76|2.99|2.51|2.11|
> |200|275|168|106|1.458|0.811|0.584|9.59|5.22|3.89|3.35|3.04|2.70|
> |300|139|114|24|1.929|0.816|0.860|10.67|6.46|5.78|5.27|4.93|4.82|
>
> **Rmk.**  #E, #V, #F are the amount of edges, vertices and faces, respectively. Usually, a mesh is considered having good quality if its elements are uniform and regular. We briefly introduce these three indicators:
> - **Aspect Ratio (AR)**. There are many versions of AR. We define it as the ratio between the maximum edge length over the minimum.
> - **Equi-Angle Skewness (EAS)**. An N-polygon's ideal angle is $\theta:=\pi (N-2)/N$. For an actual angle $\theta_{i}$, its offset is $\delta_{i}:= |\theta_{i}-\theta|/\max\{\theta,\pi-\theta\}$. And its EAS is $\max_{i}\delta_{i}$.
> - **Area Transition Ratio (ATR)**. It depicts the uniformity of element size. Let $A_i,A_j$ be the area of two neighboring elements, then its ATR is $|A_i-A_j|/\max\{A_i,A_j\}$.
>
> > **Q4.** It is unclear why the 3D magnetostatic results are not competitive (relative to other baselines) even though the model is designed to generalize naturally to 3D.
>
> This seemingly abnormal result mainly comes with different complexities of the datasets. Take Poisson equation ($\Delta u(\mathbf{x})=f(\mathbf{x}),\mathbf{x}\in\Omega;u(\mathbf{x})=g(\mathbf{x}),\mathbf{x}\in \partial\Omega$) as an example, this BVP is dominated by both source term $f(\mathbf{x})$ and boundary condition $g(\mathbf{x})$. As stated in Appendix G, the behavior of the source term $f(\mathbf{x})$ in 2D cases is more complicated in the sense that every triangle is endowed with random charges. While in 3D cases, only several embedded cubes with charge or circuits are involved. And as we know, such higher-order structure only captures local features since these higher-order features are originated by differential operators, which only reflect local behaviors. Consequently, the more the BVP is dominated by local restriction $f(\mathbf{x})$ instead of global restriction $g(\mathbf{x})$, the better performance our proposed method would achieve. In this perspective, the result that 2D cases appear better than the 3D counterparts, is consistent with both the data and the architecture.
>
> Besides, our method focuses more on how to capture local behaviors governed by $f(\mathbf{x})$ rather than the global by $g(\mathbf{x})$, due to the nature of integrated form representation. We will leave how to further combine global behaviors reflected by boundary conditions with integrated form representations for future work.
>
> > **Q5.** The comparison misses modern BVP-specific baselines such as FNO, CNO, SCOT, etc., which are more appropriate than DeepONet or Transolver (designed for time-evolving systems).
>
> Thanks for your suggestion. However, these baselines **cannot directly fit our experiment settings due to their requirement of a regular mesh**. Both FNO and CNO requires the spatial domain or its parametric space to be a regular grid (e.g., *Elliptic mesh for the airfoil problem*(p.55 fig.23)[1], Appendix 3[2]). This fact is also supported by *"state-of-the-art neural operators such as FNO, CNO, and scOT which are tailored for Cartesian grids"* [3] (p.8).
>
> Thus these baselines are improper to handle irregular meshes as shown in fig.5, which are more common in practical use. Also, it is worthwhile mentioning that involved baselines like *GNOT* also outperform earlier baselines like interpolated FNO, Geo-FNO in regular-mesh settings (Sec.4 Tab.1)[4].
>
> Still, we can understand your concerns on insufficient experiments. Therefore, we further evaluate the performance against *Geo-FNO* and *FourierType Transformer Operator*, which can handle arbitrary meshes. We will clarify why baselines like *FNO, CNO, and scOT* are not adopted in our manuscript and add extra comparisons in the updated version.
>
> | Model| 2D Eletrostatics | 2D Magnetostatics |
> |-|-|-|
> |Geo-FNO|2.017|3.289|
> |FourierType|1.553|1.986|
> |GNOT|2.064|1.142|
> |DEC-HOGNN(Ours)|**0.623**|**0.875**|
>
> Ref.
>
> [1] (CNO) Convolutional Neural Operators for robust and accurate learning of PDEs. arxiv.
>
> [2] (FNO) Fourier Neural Operator for Parametric PDE.
>
> [3] RIGNO: A Graph-based framework for robust and accurate operator learning for PDEs on arbitrary domains. arxiv.
>
> [4] GNOT:A General Neural Operator Transformer for Operator Learning.
>
> > **Q6.** *Suggestions on writing*.
> > (i) This is a very dense paper, with some parts written with excessive jargon (particularly early on), making it hard for a broader audience to parse. Key ideas (like Eq. 5) are underexplained at first reading.
> > (ii) Figures (e.g., Figure 1 and 2) are central to the understanding but require much more extensive captions and annotations to be fully self-contained.
> > (iii) The formulation of the neural operator in Eq. (5) is too abstract and hard to interpret until the reader reaches the concrete instantiation in Eq. (7). The authors could either move the motivating example earlier or introduce a simplified case (e.g., scalar Poisson) first.
>
> Thanks for your advice. For (i) and (ii), we have added more detailed explanations on these terms, equations and figures, making it more friendly to broader readers. For (iii), we have exchanged the abstract presentation of the neural operator and the concrete example properly.

---

> > ### Comment · Reviewer_1rUG · 2025-08-05
> > **Thank you for the response and additional examples**
> >
> > Thank you for the effort to address my questions.
> > I appreciate the additional comparisons with other techniques.
> > I am happy to increase the score to 5.

---

### Official Review · Reviewer_5x2r · 2025-07-03

**Clarity:** 3
**Significance:** 3
**Originality:** 3
**Rating:** 4
**Confidence:** 3

**Summary:**

solve boundary value PDE problems. The authors incorporate discrete and finite element exterior calculus into GNNs to deal with geometrical relationships between differential forms (e.g., charge density and electric field). They also derived the physics-informed loss to measure the consistency of the output against the considered physical law. In addition, the authors provided the universal approximation theorem for Poisson problems, justifying the use of the model for electromagnetic phenomena. The experimental results suggest that the proposed method has a high capacity to predict electromagnetic phenomena for 2D and 3D settings.

**Questions:**

* In the experiments, how much was the error of PDE constraints, e.g., divergence-free condition?
* What would be necessary to extend the method for other type of PDEs, e.g., Navier–Stokes equations?

**Ethical Concerns:**

["NO or VERY MINOR ethics concerns only"]

**Final Justification:**

Although most of the concerns are addressed by the authors, the evaluation of the speed-accuracy tradeoff compared to classical solvers, which is the most essential one, is yet to be addressed. Therefore, I keep the original score.

**Limitations:**

yes

**Quality:**

4

**Strengths And Weaknesses:**

Strengths:
* The writing is well-construcgted and easy to follow. The authors reveal the connection of the present method to existing ones from the viewpoint of DEC (discrete exterior calculus), clarifying the position of the research.
* The derivation of physics-informed loss for GNN is useful. Thanks to the DEC formulation, a certain physical constraint turned out to be the sum of integration, which is easy to compute.
* The method is demonstrated to have high accuracy compared to the considered baseline models, showing the expressibility of the model.

Weakness:
* The relation to existing exterior-calculus-based or simplicial-complex-based methods is not clearly stated. For instance, differences and advantages against research on cellular complexes [Alain+ ICML 2024] and simplicial complexes [Ebli+ NeurIPS Workshop 2020] could be stated, and possibly, compared experimentally.
* There is no evaluation of computation time. Since the classical solvers have high accuracy (with possibly a lot of computation resources), the evaluation of the accuracy alone is not enough. Therefore, the reviewer recommends evaluating the speed-accuracy tradeoff with changing spatiotemporal resolution (at least for classical solvers).

Minor points:
* Simplex complex (l.63 p. 2) could be called simplicial complex instead.
* Since x is not used in the PDE, $x \in \Omega$ in Equation 6 is confusing. Either using x in the PDE or writing $\mathrm{in} \ \Omega$ would be preferable.

---

> ### Author Rebuttal · Authors · 2025-07-29
>
> We are grateful for your thoughtful review and constructive suggestions. All your comments have been carefully addressed in the point-by-point responses below.
>
> > **Q1**. For instance, differences and advantages against research on cellular complexes [Alain+ ICML 2024] and simplicial complexes [Ebli+ NeurIPS Workshop 2020] could be stated, and possibly, compared experimentally.
>
> Thanks for pointing out these important works on incorporating higher-order topology. A crucial observation is that one can well define the neighborhood of a $k$-cell as a set of $(k-1),k$ or $(k+1)$-cells, respectively. Since lots of canonical results of graphs are based on a certain type of adjacency, one can generalize the usual graph Laplacian to higher-order Laplacians, based on which *Ebli et al.* proposed a higher-order GCN while *Alain et al.* derived a higher-order graph Gaussian process. **The message passing mechanism in our paper actually covers Ebli et al.'s work**. These methods take advantage of multi-body interactions in different scales in a topological structure since a $k$-cell can be interpreted as the result of the interactions among $k$ individuals (nodes).
>
> Our approach is a step beyond merely leveraging topological inductive bias. Indeed, by connecting higher-order features with differential forms, it **paves the way for further leveraging geometry inductive bias** since forms are more than commonplace in differential geometry. By encoding input data as forms and building their relations properly, our method is able to turn differential operators into higher-order feature interactions. In all, our approach is **an attempt to open the door towards the geometry world**.
>
> > **Q2**. Since the classical solvers have high accuracy (with possibly a lot of computation resources), the evaluation of the accuracy alone is not enough. Therefore, the reviewer recommends evaluating the speed-accuracy tradeoff with changing spatiotemporal resolution (at least for classical solvers).
>
> Thanks for your suggestion. The neural operators often outperform classical solvers by **several orders of magnitude** in terms of speed and thus the time cost comparison is less talked about. As stated in FNO[1] (Sec. 1), "*On a 256×256 grid, the Fourier neural operator has an inference time of only **0.005s** compared to the **2.2s** of the pseudo-spectral method used to solve Navier-Stokes.*"
>
> **In our setting, the Ansys 3D electrostatics solver takes 14 seconds in total  and 427MB memory for adaptive meshing to reach 0.1% energy loss, let alone time-consuming work on preliminary mesh checking.** Conversely, most prevailing neural operators are on the milisecond level. Furthermore, our experiments mainly concentrate on the effectiveness of differential-form neural representations, and thus only time-independent PDEs are considered, i.e., it **does not involve spatiotemporal resolution**.
>
> But still, to validate our approach on time efficiency, we compare our methods with neural baselines. We kindly invite you to refer to the response to **Reviewer RTKj 's Q1** where a comprehensive comparison on computation cost against neural operators is presented.
>
> > **Q3**. In the experiments, how much was the error of PDE constraints, e.g., divergence-free condition?
>
> Let us further clarify some technical details on this topic.
>
> In the magnetostatics experiments, we optimize the divergence-free loss $\mathcal L_{\text{{div}}}$ weighted by face area (in 2D cases) or cell volume (in 3D cases), and the data loss $\mathcal L_{\text{{data}}}$ simultaneously. Note that divergence-free loss **should be viewed as a requirement instead of a regularization term**. Otherwise, it may overfit the data at the vertices and the complete field recovered via interpolation can thereby go against physics laws.
>
> A further experiment is conducted: we set the total loss $\mathcal L:=\mathcal L_{\text{{data}}}+10^2\mathcal L_{\text{{div}}}$ and then observe the training curves with and without this extra divergence-free condition.
>
> We define the ratio $R:=10^2\mathcal L_{\text{{div}}}/\mathcal L$. The higher $R$ is, the more likely that the model overfits the data at the vertices and breaks the physics law. And also, we study whether this extra term would have a negative impact on fitting given data points by observing the validation data loss $\mathcal L_{\text{{data}}}$. The table below shows two validation curves within 500 epochs. The result shows that **merely fitting the data points is likely to violate the divergence-free condition**.
>
> | Train with $\mathcal L_{\text{{div}}}$ | Item                            | 0     | 100    | 200    | 300    | 400    | 500    |
> | -------------------------------------- | ------------------------------- | ----- | ------ | ------ | ------ | ------ | ------ |
> |                                        | $R$                             | 1.05% | 51.76% | 74.89% | 85.88% | 90.09% | 92.47% |
> |                                        | $\mathcal L_{\text{{data}}}$    | 9.44  | 4.39   | 2.97   | 2.26   | 1.81   | 1.54   |
> |                                        | $10^2\mathcal L_{\text{{div}}}$ | 0.10  | 4.71   | 8.86   | 13.74  | 16.45  | 18.91  |
> | √                                      | $R$                             | 0.73% | 1.56%  | 4.00%  | 5.23%  | 6.67%  | 8.98%  |
> | √                                      | $\mathcal L_{\text{{data}}}$    | 9.47  | 5.06   | 3.60   | 2.90   | 2.52   | 2.23   |
> | √                                      | $10^2\mathcal L_{\text{{div}}}$ | 0.07  | 0.08   | 0.15   | 0.16   | 0.18   | 0.22   |
>
> Thus the conclusion is: **without this divergence-free loss term, the model is prone to overfit the data at sampled points, hence, greatly violating the divergence-free condition.** Though sometimes fitting slower at given data points, some properties of the field are preserved, like *divergence-free*, coinciding with the insights of preserving symmetries in *Lagrangian Network* and *Hamiltonian Network*.
>
> > **Q4**.  What would be necessary to extend the method for other type of PDEs, e.g., Navier–Stokes equations?
>
> As stated, our current method can cover various linear PDEs including common operators like $\text{grad, curl, div}$. While some geometric neural representations are needed to achieve non-linear PDEs. Below, we briefly demonstrate some possible technical approaches for solving Navier-Stokes Equation (NSE) and leave them for future work.
> - The viscosity $\mu\Delta \mathbf{u}$ term in NSE can be decomposed into $\mu \text{grad } (\text{div }\mathbf{u})$ and $\mu\text{curl }(\text{curl } \mathbf{u})$. In 2D cases, the integration of $\mathbf{u}\cdot \mathbf{n}$ on an edge $e$ ($\mathbf{n}$ is the normal of a primal edge and also the tangent of its dual counterpart $\star e$) can be interpreted as the flux through $e$ and thus a 1-form on $\star e$. This again represents $\text{div }\mathbf{u}$ properly via the *edge coboundary* in HOGNN, while $\text{grad}$ can be implemented on the dual graph based on graph calculus. The other term $\mu\text{curl }(\text{curl } \mathbf{u})$ can be likewise handled.
> - Directly handling the non-linear term $\mathbf{u}\cdot \nabla \mathbf{u}$ may be difficult. However, this term is associated with Lie derivative $\mathcal L_{X}$, which is more profound in differential geometry and allows generalization to non-Euclidean spaces. There already exist some preliminary implementations of Lie derivative and vector fields onward. Nevertheless, how to represent them effectively for neural networks remains open and is worthwhile to explore in the future.
>
> > **Q5**. On some minor issues.
> > (i) Simplex complex (l.63 p. 2) could be called simplicial complex instead.
> > (ii) Since x is not used in the PDE, $x \in \Omega$ in Equation 6 is confusing. Either using x in the PDE or writing in $\Omega$ would be preferable.
>
> Thanks for your correction. We have modified both in our manuscript. For (ii), it is indeed a bit confusing though $\mathbf{x}$ is implicitly used in the fields.

---

> > ### Comment · Reviewer_5x2r · 2025-08-04
> >
> > Thank you for the response.
> >
> > Q2:
> > Evaluation of the speed is not enough. We have to evaluate the speed-accuracy tradeoff. Because machine learning is not perfectly accurate, the comparison based only on computation time is not complete. If we are allowed to reduce accuracy, we can reduce spatial resolution or increase the convergence threshold on the classical solvers.
> >
> > That's why I recommend evaluating the speed-accuracy tradeoff both for machine learning and classical solvers. I expect to see a fair comparison, similar to Figure 4 of [Horie and Mitsume, Physics-Embedded Neural Networks: Graph Neural PDE Solvers with Mixed Boundary Conditions, NeurIPS 2022], as we are aware that machine learning methods often struggle to improve the speed-accuracy tradeoff compared to classical solvers in a fair manner.
> >
> > Q3
> > I see that the divergence-free loss term is important for the setting in the paper. But I guess we can achieve (at least mathematically) perfect divergence-free when we use the vector potential. Why is the vector potential not used for the experiments?

---

> ### Author Response · Authors · 2025-08-05
>
> Thanks for your further explanation.
>
> > **Q6.** Evaluation of the speed is not enough. We have to evaluate the speed-accuracy tradeoff. Because machine learning is not perfectly accurate, the comparison based only on computation time is not complete. If we are allowed to reduce accuracy, we can reduce spatial resolution or increase the convergence threshold on the classical solvers.
>
> Thanks for your suggestion. It is indeed important to evaluate the speed-accuracy tradeoff of different methods. But due to limited time in rebuttal session, we cannot offer a comprehensive result by changing many factors like grid resolution and various model hyperparameters. Here we only choose some competitive baselines and change model hyperparameters as [1] did. More results would be updated in the next version of our manuscript.
>
> By changing the hyperparameters of the model, we evaluate the computation overhead of a model by recording its computation time on one core of a CPU following [1]. It can be seen that this trade-off is more complicated than the one in numerical methods. **More computation overhead (lower speed) does not always come with faster convergence and higher accuracy** since the performance is also constrained by dataset amount, the sparsity of model parameters, overfitting and mesh regularity (some appear in numerical methods).
>
> This pheonomenon on the 2D magnetostatics benchmark (see tables below) also aligns with Fig. 4 in [1] where data points on the MSE-time plane are not necessary to be strongly related (PENN and OpenFOAM are the most salient). Also, as in Table 5[1] (row 2 and 4; row 3 and 5), more computation can sometimes bring about performance drop and training difficulty. **In summary, under finite data and other constraints, neural architectures exhibit distinct computational requirement intervals. In the tradeoff against accuracy, speed is a key factor but not all.**
>
> Tab 1: Models with different computation cost on the 2D magnetostatics benchmark.
> |Model|Configuration|ComputationTime(s)|TestLoss|
> |-|-|-|-|
> |GNOT|32|1.60|6.128|
> |GNOT|64|2.42|3.362|
> |GNOT|128|4.75|2.406|
> |GNOT|256|12.32|4.189|
> |DEC-HOGNN(Ours)|32|3.58|2.959|
> |DEC-HOGNN(Ours)|64|5.78|2.831|
> |DEC-HOGNN(Ours)|128|11.68|**1.573**|
> |DEC-HOGNN(Ours)|192|18.82|2.437|
> |Transolver|(128,8,32)|2.66|3.312|
> |Transolver|(256,8,32)|4.59|2.829|
> |Transolver|(512,8,32)|10.11|3.888|
> |Transolver|(512,16,64)|10.93|3.758|
> |GalerkinType|32|2.99|2.649|
> |GalerkinType|48|5.36|2.899|
> |GalerkinType|64|11.81|1.932|
> |GalerkinType|128|62.77|2.492|
>
> Tab 2: Best performance with different levels of computation.
>
> | Time       | [0,5)     | [5,10)           | [10,15)            | [15,∞)         |
> | ---------- | --------- | ---------------- | ------------------ | -------------- |
> | Best Model | GNOT(128) | GalerkinType(64) | **DEC-HOGNN(128)** | DEC-HOGNN(192) |
> | Loss       | 2.406     | 1.932            | **1.573**          | 2.437          |
>
> **Rmk.** The configurations of GNOT, DEC-HOGNN and GalerkinType only involve latent space dimension. As for Transolver, $(x, y, z)$ means it has $x$-dimensional hidden features, $y$ heads and $z$ slices.
>
> Ref.
>
> [1] Horie and Mitsume. Physics-Embedded Neural Networks: Graph Neural PDE Solvers with Mixed Boundary Conditions.
>
> > **Q7.** Why is the vector potential not used for the experiments?
>
> By introducing a vector potential $\mathbf A$, it can indeed guarantee the divergence condition. However, it has several flaws.
> - First, one must estimate its curl since $\mathbf B=\nabla\times\mathbf A$. But an accurate computation is infeasible because only finite observations on the graph vertices are available. Though one may obtain an approximation via interpolation-based methods like reproducing Hilbert kernel or machine-learning approaches, in whichever way, **the divergence-free condition is still violated**.
> - Second, if we directly use a network to represent this potential (that is, take in a 3D coordinate and output the potential $\mathbf A$), the notorious issues in PINN arise:
> 	- It is usually hard to converge since **higher-order derivative behaviors are harder to control** during learning.
> 	- Also, in our benchmarks, medium are different in diferent parts of the domain, indicating that such a potential is **not continuous** around these borders.
> 	- Some **irregularities** of this potential can also arise in some scenarios (e.g., around a current-carrying thin wire).
> 	- Furthermore, such a vector potential is **not unique** unless being restricted by some gauges, like divergence-free condition (Coulomb gauge). Whether this extra degree of freedom would give rise to learning difficulties is unknown.
> In all, by introducing integrated forms, our proposed approach avoids directly handling higher-order derivatives of quantities like vector potential which has irregular behaviors and is usually tough to learn.

---

> > ### Comment · Reviewer_5x2r · 2025-08-07
> >
> > Thank you for the answer. I see the complexity behind the vector potential.
> >
> > Although I agree that the time for rebuttal is quite limiting, I believe the authors should clearly present the benefit of using machine learning over classical solvers (and other machine learning models, which is addressed in the last reply) in a quantitative way. Therefore, I will keep my score as is.
> >
> > Nevertheless, I acknowledge the mathematical solidity of the work.

---

### Decision · Program_Chairs · 2025-09-17

**Decision:**

Accept (spotlight)

**Comment:**

The paper presents a GNN-framework for solving partial differential equations that incorporates higher-order interactions based on discrete and finite element exterior calculus. The method outperforms existing neural operators and  provies theoretical guarantees of universal approximation for certain voundary value problem classes. Reviewers noted that the methodology is technically sound, highly innovative, and appreciated the rigorous theoretical support. They also commented that the work opens up a new direction for learning-based PDE solvers, that the physics-informed loss is elegant, and they appreciated the improved accuracy compared to baselines. On the other hand, reviewers raised concerns about clarifying the relationship to existing methods based on exterior calculus or simplicial complexes, the clarity of the writing, the relatively limited number of experimental results, and the omission of certain baselines (e.g., FNO, CNO, SCOT).

The authors responded to the reviewer concerns with a rebuttal, which provided additional experimental evaluation, clarified why certain baselines were not included (e.g., due to their requirement for a regular grid), added an assessment of computational costs, and replied to a number of other technical questions. The reviewers agreed that all or most of their concerns were addressed by the rebuttal, and there was consensus to accept the paper.

I recommend the paper be accepted, and I also note that it stands out from other papers because of the thoroughness of the work, the technical novelty of the approach, and the potential for broad impact in the area of neural operators and physics-informed neural networks.